# Where There’s a Will, There’s a Way? Social and Mental Forces of Successful Adaptation of Immigrant Children in Young Adulthood

**DOI:** 10.3390/ijerph19116433

**Published:** 2022-05-25

**Authors:** Jerf W. K. Yeung, Hui-Fang Chen, Zhuoni Zhang, Andrew Yiu Tsang Low, Herman H. M. Lo

**Affiliations:** 1Department of Social and Behavioral Sciences, City University of Hong Kong, Kowloon, Hong Kong, China; hfchen@cityu.edu.hk (H.-F.C.); yiutlow@cityu.edu.hk (A.Y.T.L.); 2Urban Governance and Design Thrust, Society Hub, The Hong Kong University of Science and Technology (Guangzhou), Guangzhou 511458, China; znzhang@ust.hk; 3Division of Social Science, School of Humanities & Social Science, The Hong Kong University of Science and Technology, Hong Kong, China; 4Division of Public Policy, Interdisciplinary Programs Office, The Hong Kong University of Science and Technology, Hong Kong, China; 5Department of Applied Social Sciences, The Hong Kong Polytechnic University, Hong Kong, China; herman.lo@polyu.edu.hk

**Keywords:** children of immigrants, successful adaptation, social forces, mental forces

## Abstract

Although the twenty-first century is deemed as a new era of globalization, waves of immigration continue, due to disparities between politically and economically unstable regions and Western democratized and developed countries. Immigration research has therefore reignited its attention on the successful adaptation of immigrants’ offspring, which has profound implications for Western immigrant-receiving countries, as well as worldwide stability. Although immigration research mainly informed by the conventional assimilation theory and/or segmented assimilation perspective accentuates the importance of structural factors, termed as social forces here, in relation to immigrant children’s successful adaptation in adolescence, an argument of determinism and tenability keeps on and the contribution of human mental resources and determination, termed as mental forces here, in shaping life trajectories of immigrant children should be not ignored. For this, with a representative sample of 3344 immigrant children from the Children of Immigrants Longitudinal Study (CILS), we examined and compared both the effects of social and mental forces measured in adolescence of immigrant children on their multiple adaptation outcomes in terms of college graduation, engagement in postgraduate study, and first and current job attainments in young adulthood with a Bayesian multilevel modeling framework. The results found that both social forces of segmented assimilation theory and mental forces of immigrant children in adolescence were significantly predictive of immigrant children’s successful adaptation in young adulthood (OR = 1.088–2.959 and β = 0.050–0.639 for social forces; OR = 11.290–18.119 and β = 0.293–0.297 for mental forces), in which, although the latter showed stronger effects than the former, the effects of mental forces on adaptation of immigrant children were conditionally shaped by the contexts of the social forces informed by segmented assimilation theory. The findings of the current study highlight the significance of the organism–environment interaction perspective on immigration research and provide an insight to consider a context-driven response thesis proposed.

## 1. Introduction

The pendulum of immigration research has recently swung back to the successful adaptation of children of immigrants. The reemergence of research interest in immigrants and the future of their offspring has been spurred by the global refugee crisis [1,2]. Specifically, successful integration to host societies by immigrants and their children, a process referring to economic mobility and social inclusion, has profound implications for Western immigrant-receiving countries, such as legal order, social stability, economic and labor development, cultural diversity, and general societal well-being [3,4]. Although prior research on adaptation of immigrant children has covered various topics ranging from high-school dropout, educational performance, and social and emotional adjustment [5,6,7], most limited their investigation of the development of immigrant children within adolescence only. Empirically, little research has extended the time frame of immigrant youths from early adolescence to young adulthood to examine the process of their successful adaptation [3,4,8]. Another limitation of prior research on successful adaptation of immigrant children is its predominant focus on the effects of structural and societal factors, termed as social forces here, on the development of immigrant children [4,9,10]. In fact, human mental strength and psychological determination, referred to as mental forces here, may play a critical role in shaping immigrant children’s later life trajectories and adaptation but have received little empirical attention in the immigration literature [11,12]. Furthermore, although existing immigration studies based on the conventional assimilation perspective and/or the dominating theory of segmented assimilation have confirmed the importance of structural and societal factors in contribution to immigrant children’s adaptation outcomes, few have simultaneously examined relevant structural and societal effects at both individual and contextual levels [3,13]. Apparently, investigating successful adaptation of immigrant children independent of the compositional effects derived from their socializing contexts, such as the school socialization environment, may incur biased support of the theorizing relationships [14].

In this study, we intended to construct a causal relationship investigating the effects of social and mental forces and their interplay on immigrant children’s successful adaptation in terms of college graduation, enrolment in postgraduate study, and first and current occupational attainments from early adolescence to young adulthood. The present investigation surpasses prior immigration research. First, it covers an extended developmental period of immigrant children from early adolescence to young adulthood stretching a 10-year time span and concurrently examines multiple adaptation outcomes. Second, the effects of social forces in comparison to that of mental forces on successful adaptation of immigrant children are empirically examined to broaden our knowledge on the developmental trajectories of this population. Third, due to social and cultural segregation in different ethnicities, this study decomposes the clustering nature of their socialization contexts by concomitantly investigating both individual- and contextual-level influences of social and mental forces on immigrant children’s adaptation outcomes in young adulthood [13]. Finally, and most important, this study analyzes the interplay between social and mental forces to explore their possible dynamic and conditional impacts on successful adaptation of immigrant children in young adulthood.

## 2. Theoretical Framework of Successful Adaptation of Immigrant Children

Researching successful adaptation of immigrant children has both scholarly and practical values. Unlike their parents, immigrant children are citizens and entitled to claim their rights by birth or naturalization [5,15]. In 2019, 25.8% of 69 million children under 18 in the United States were from immigrant families, constituting 17 million people [16]. Compared to their US-born native counterparts, immigrant children generally grow up in disadvantaged environments and face various structural and contextual difficulties for their success in adulthood [4,15,17]. Although these children live in comparatively disadvantaged and difficult conditions, some of them are still capable of attaining higher academic and social achievement [15,18]. Therefore, it is pivotal for researchers to investigate how and under what circumstances immigrant children can successfully adapt in young adulthood [3,8]. Rumbaut [19] described immigrant children’s successful adaptation in young adulthood as a turning point to “‘knife off’ the person’s past from the present and serve as catalysts for long-term behavioral change by restructuring routine activities and life course pathways (p. 1042)”. In the following sections, we present a theoretical framework to construct and delineate the causal effects of social and mental forces and their interplay on successful adaptation of immigrant children in young adulthood. We then go to test the study relationships by data of a large and representative sample from the Children of Immigrants Longitudinal Study (CILS).

### 2.1. Hypothesis of Assimilation Theories and Adaptation of Immigrant Children

Prior immigration studies, mainly informed by the conventional assimilation theory and/or segmented assimilation perspective, accentuate the structural and societal factors in relation to successful adaptation of immigrant children. These two assimilation theories highlight the significance of social mobility and convergence into the mainstream culture of host societies as the final end, resonant with what Gordon [20] claimed as the White Protestant middle-class in the United States of America. The distinct differences between the two perspectives are that the former stresses the concept of “straight-line” convergence and assimilation as “an integral part of the process of upward mobility” [9] (p. 109), and the latter emphasizes ethnic divergences in the process of assimilation and “several distinct paths of adaptation” prominently appearing by the socialization impacts of family and proximal environmental factors [3] (p. 737). Regarding the conventional assimilation theory, although scholars have once queried its applicability in contemporary immigration waves with a dubitable question marked by Glazer [21] that “(i)s assimilation dead?”, recent research reassured its external validity in predicting adaptation of immigrants [9,10,22]. Segmented assimilation theory, which is mainly derived from the critiques of the conventional assimilation perspective, has afterward become the mainstream theoretical precept of immigrant adaptation enshrined in the literature [4,15]. Despite its effective predictive power of development of immigrant children, however, some researchers have recently challenged its assertion of variegated and segmented assimilations [10,22]. Nevertheless, the valuable contributions of these two assimilation theories, both rooted in structuralist and functionalist traditions, are their systematic deliberation of different structural and societal factors in relation to the adaptation of immigrant children in Western capitalist host societies.

Apparently, the conventional assimilation perspective reckons that the host society defines and controls most productive and critical resources for socioeconomic advancement, and the process of full assimilation is inextricable and needed [9,23]. Hence, scholars of the conventional assimilation perspective stress the importance of specific structural factors for successful adaptation of immigrant children, which include immigrant generations, length of residence, use of language, native friendship, and residential locality [23]. Although scholarly doubt was once cast on its applicability in contemporary immigration waves, researchers have recently supported the external validity of conventional assimilation theory in prediction of adaptation of immigrants through general convergence to the outcomes of their native White counterparts [10,22,24]. Segmented assimilation theory, in contrast, insists that the United States, similar to most Western capitalist nations, is a highly stratified and unequal society [3,15]. Therefore, immigrant children will assimilate into different “segments” of society [4,15]. Some are upwardly assimilated, such as the assumption of conventional assimilation perspective posits, and some are downwardly assimilated into the urban underclass, leading a life of poverty and marginalization, and the remaining may integrate into the mainstream host society by preserving their ethnic culture and doing economic integration, thereby finally setting to diverse life trajectories [4,5,15]. Accordingly, segmented assimilation theorists deem that family composition, family socioeconomic status (SES), and mode of incorporation, or called reception mode, are the most critical structural contributors relating to successful adaptation of immigrant children [2,3,25]. Thereby, this perspective posits that immigrant children of two-biological-parent families (two-parent family hereafter for simplicity), higher family SES, and positive, or at least neutral, reception will do better than their counterparts from broken and poor family backgrounds and negative reception [15,26]. Evidently, the effects of family composition, family SES, and incorporation mode on the adaptation process of immigrant children appeared more influential than those of structural factors specified by the conventional assimilation theory [13,22,27]. In fact, children, especially during adolescence, are in the formative years of critical development. They are more susceptible to the influences of family socialization and immediate receiving contexts [28,29]. Therefore, family background and receiving context are the proximal and nurturing socialization agents to formulate the most profound cognitive and interpersonal experiences of immigrant children for their social and educational development, such as language use, peer interactions, educational attitudes, behavioral choices, and career goals, which later lead to their different adjustment and life trajectories [13,28,30]. Thus, compared to the structural factors proposed by the conventional assimilation theory, we expect that family composition, family SES, and the incorporation mode specified by the segmented assimilation perspective would be more strongly predictive of the successful adaptation of immigrant children in young adulthood (hypothesis 1).

### 2.2. School Context and Adaptation of Immigrant Children

Immigrant children are generally students studying in school and spend much of their waking time in the school environment [22,31]. It is reasonable to believe that the school context may have inextricable compositional impacts on the adaptation and development of immigrant children. Through interaction with classmates, teachers, and other school personnel, as well as observations of the school environment, immigrant children may cultivate and develop educational attitudes, life orientations, and behavioral choices that may, in turn, shape their future life chances and trajectories [32,33]. Accumulated empirical research has confirmed school composition as a key socialization domain influencing children’s future academic and occupational success beyond individual-level characteristics [22,34]. This is consonant with the eco-developmental theory that youth development is simultaneously susceptible to both individual- and contextual-level influences [28,35], in which the compositional effects of the school environment are directly relevant to their future adaptation and success [34,36]. Nevertheless, with few exceptions, prior immigration research has seldom examined the compositional effects of school environment on the successful adaptation of immigrant children in young adulthood. For example, Hermansen and Birkelund [37] examined only limited structural factors in relation to immigrant children’s high school completion and graduation GPA. In the present study, we intend to analyze both individual- and school-level effects of family composition, family SES, and incorporation mode highlighted by the segmented assimilation theory on immigrant children’s adaptation outcomes of college graduation, enrolment in postgraduate study, and first and current occupational attainments in young adulthood, while adjusting for pertinent school-level characteristics considered to affect immigrant children’s developmental trajectories. These characteristics include school location, school type, minority-receiving school, and aggregated standardized test scores of the school [22,36].

Conceivably, the aggregation of family composition, family SES, and incorporation mode at the school level connotes a school’s capacity to recruit quality teachers, drawing parental school engagement, acquiring resources, and providing learning experiences to students [33,38]. If a school is of higher SES position, more two-parent families, and students with positive reception, it is believed that this school can have more favorable learning environment and educational experiences for the better adaptation of immigrant children. Moreover, the literature reports that immigrant children in urban, public, and minority-receiving schools fare more poorly than their counterparts in rural, private, and native schools [22,33]. Further, students in schools of higher average academic achievement at school level tend to have better educational performance [33,39]. Therefore, we expect that family composition, family SES, and incorporation mode specified by segmented assimilation theory would be simultaneously predictive of immigrant children’s successful adaptation in young adulthood at both the individual and school levels (hypothesis 2).

### 2.3. Mental Forces and Adaptation of Immigrant Children

Human development is a dynamic and interactive process rather than static and unidirectional in which intrapersonal character and mental resources are in interplay with external structural factors synchronously, leading to different developmental trajectories [40,41]. However, neither the conventional assimilation theory nor the segmented assimilation perspective acknowledges the importance of immigrant children’s mental strengths and psychological resources as possible mental forces contributing to their later successful adaptation. Informed by the organism–environment interaction thesis [42], individual development of immigrant children will not be equally affected by the same or similar contextual and structural conditions, and immigrant children may progress toward differential trajectories depending on their mental resources and intrapersonal character [28,43]. Like most general youths, immigrant children are in a critical period of physical, cognitive, psychological, and behavioral development. Their self-esteem and sense of hopeful future are important mental strengths and resources pertinently relating to their resolution and resilience to help them overcome structural obstacles and difficulties for better development [44,45]. Self-esteem is defined as an overall concept of self-worth, self-acceptance, and confidence for forming a crucial cognitive basis to affect immigrant children’s values, goals, and behavioral choices related to their future development [44,46]. A sense of hopeful future, or positive future expectation, is both a cognitive and motivational basis for immigrant children’s capability of planning their future and setting actions to achieve goals set as valuable [45,47].

Research has shown that higher self-esteem and positive future expectation are related to the better adaptation of youths [44,48]. Related studies have reported that self-esteem is related to fewer antisocial behaviors, and maladaptation and emotional problems, as well as better interpersonal relationships, and academic achievement among immigrant and non-immigrant children [33,49]. In addition, future expectation, involving the ability to plan for the future and set actions to attain planned goals, is conducive to the educational and occupational success of immigrant children. Adelabu [50] found that positive future orientation significantly predicted higher educational achievements among a sample of African American adolescents. Another study by Portes, Fernandez-Kelly and Haller [4] reported that educational future expectation of immigrant children negatively predicted their downward assimilation. Nevertheless, although self-esteem and future expectation demonstrate as possible and important mental forces to shape the developmental trajectories of immigrant children, little research has investigated their effects on the successful adaption of immigrant children while, at the same time, adjusting and comparing the structural effects, which are the social forces, informed by the conventional assimilation theory and segmented assimilation perspective. Furthermore, prior research tended to treat self-esteem and the future orientation of immigrant and non-immigrant children as manifest variables in relation to their adaptation [3,4,44], which, in fact, are latent [51,52]. In the terminology of methodologists, if a latent construct is treated as manifest instead, unobserved measurement errors occur, and inaccurate study relationships are likely obtained. Moreover, prior research found that children’s self-esteem and future expectation were highly correlated and mutually reinforced each other [44,53]. The possible mechanism is revealed by the self-enhancement theory that the “future self” is an extension of the “current self” [44,54], in which youths’ current worth, values, and self-image may translate to their hopes, goals, and future plans that then help sustain their current self-concept positively [29,54]. In a recent study investigating the relationship between youths’ self-perception and future expectations, Verdugo, Freire and Sanchez-Sandoval [53] corroborated that the correlation between self-esteem and future expectation was r = 0.51, *p* < 0.001. Another longitudinal study by Jackman and MacPhee [44] also reported that youths’ self-esteem significantly correlated with their future orientation concurrently (time 1), r = 0.42, *p* < 0.001, and prospectively (time 2), r = 0.46, *p* < 0.001.

In this study, we treat the mental forces of immigrant children as a second-order latent construct by loading self-esteem and educational future expectation as the first-order latent factors and converge them on the second-order mental forces of immigrant children. Accordingly, we expect that the mental forces of immigrant children would be positively and robustly predictive of their successful adaptation later in young adulthood, even adjusting for the structural effects proposed by conventional assimilation theory and segmented assimilation perspective (hypothesis 3). Moreover, we expect that the effects of mental forces of immigrant children on their successful adaptation would be significantly stronger than the structural effects of family composition, family SES, and incorporation mode informed by segmented assimilation theory (hypothesis 4). Validly, immigrants and their children in Western developed and industrialized countries mainly came from economically and politically underdeveloped regions to seek “opportunities” of upward mobility and life transformation, even at the cost of contextual disadvantages and discrimination [3,15]. Therefore, we assume that the effects of immigrant children’s mental forces on some adaptation outcomes would be more resilient and dynamic than on others, depending on the contextual nature of family composition, family SES, and incorporation mode. This is consonant with what Portes and Hao [33] mentioned: “(d)espite such handicaps, the majority of Mexican-American students still manage to graduate from high school and a minority even moves up to college. That achievement reflects the resilience of individual determination despite adverse external circumstances (p. 11927)”. We call these dynamic effects of immigrant children’s mental forces in interplay with contextual influences as a proposition of the context-driven resilient response and describe it below.

### 2.4. Context-Driven Resilient Response of Immigrant Children’s Mental Forces and Their Adaptation

Although the organism–environment interaction perspective signifies human development and life trajectories are the result of one’s psychological and contextual interplay, it does not explicate how and under what contextual circumstances mental forces may function differently and conditionally. In this study, we propose a thesis of context-driven resilient response to posit that human determination and mental forces are more stimulated in certain contexts and/or for certain outcomes. Specifically, if the disadvantaged situations can be revamped and reversible and/or the outcomes are attainable by human efforts and individual perseverance, the effects of mental forces would function more resilient and responsive in these contexts and for these outcomes. First, the main purpose of immigrants and their children coming from politically unstable and economically underdeveloped regions to Western industrialized and economically developed countries is for upward social mobility; hence, obtaining a first college degree in their host countries is most fundamental for immigrant children to change their life trajectories and achieve upward mobility. This corresponds to what Portes, Fernandez-Kelly and Haller [4] wrote: “(w)ithout the costly and time-consuming achievement of a university degree, such dreams are likely to remain beyond reach (p. 1081)”. Therefore, we believed that immigrant children of non-two-parent families, lower family SES, and negative reception conditions need to further harbor and evince stronger confidence, determination, and strength to overcome more structural difficulties and obstacles for college graduation [26,55]. We expect that the effects of mental forces on college graduation would be more salient for immigrant children from non-two-parent families, lower family SES, and negative reception conditions as compared to their counterparts of two-parent families, higher family SES, and positive/neutral reception status (Hypothesis 5).

Similarly, engagement in postgraduate study in young adulthood bears tangible and symbolic meaning for immigrant children in non-two-parent families, lower family SES, and negative reception conditions. They may think postgraduate education is a valuable opportunity to “equalize” their humble social positions and is an important asset to achieve upward social mobility [51,55]. This is especially important for immigrant children of negative reception, because postgraduate education and the acquisition of a higher degree can help efface their discriminative blemish derived from their negative reception [55,56]. Thus, we expect that the effects of mental forces of immigrant children on their engagement in postgraduate study would be more salient for immigrant children in non-two-parent families, lower family SES, and negative reception conditions than for their counterparts in two-parent families, higher family SES, and positive/neutral reception status (hypothesis 6).

In other ways, the nature of labor market is very different from the realm of higher education, in which it is difficult to preclude favoritism or even tacit discrimination in labor market compared to higher education [34,55]. Manifestly, acquisition of a good first job requires not only resilient mental forces and educational credentials but also family resources, interpersonal connections, and a supportive social network [15,56]. Therefore, due to their relatively weak family resources and social connections, immigrant children in non-two-parent families, lower family SES, and negative reception conditions are in a less favorable position to gain better first occupation attainment as compared to their counterparts of two-parent families, higher family SES, and positive/neutral reception [55,56]. Thus, we believed that different effects of mental forces on the first occupational attainment in their young adulthood are confounded and counteracted to be indistinguishable for immigrant children in the different contexts of family composition (non-two-parent vs. two-parent), family SES (low vs. high), and incorporation mode (negative vs. positive/neutral). Therefore, we expect that the effects of mental forces on first occupational attainment would be comparable among immigrant children in the different contexts of family composition, family SES, and incorporation mode (hypothesis 7).

However, human and social capital and work experience accumulate continuously and persistently after graduation, and hence, the opportunity of first occupation would help immigrant children in non-two-parent families, and lower family SES contexts compensate for prior family and social disadvantages [56,57]. Thus, we expect that the effects of mental forces on current occupational attainment would be more salient for immigrant children in the contexts of non-two-parent families and lower family SES conditions than their counterparts of two-parent families and higher family SES status (hypothesis 8). However, due to negative reception derived from social prejudices and ideological biases entrenching with deep cultural and historical roots [3,15], the accumulation of human and social capital and work experience may not easily help immigrant children of negative reception to eliminate their unfavorable treatment in the labor market, a phenomenon we call as persistent countervailing effects of cultural and historical origin. Therefore, we expect that the effects of mental forces on the current occupational attainment would still be confounded to be indistinguishable among immigrant children in the negative and positive/neutral reception contexts (hypothesis 9). Table 1 summarizes the dynamic and conditional effects of immigrant children’s mental forces on different adaptation outcomes across the contexts of family composition, family SES, and incorporation mode, in which H connotes the higher effects of immigrant children’s mental forces in the specific contexts of family composition, family SES, and incorporation mode compared to their counterparts in the opposite contexts, where L connotes the lower effects of mental forces. If comparable effects of immigrant children’s mental forces in the two relative contexts are expected, N is used to connote their parallels.

## 3. Data and Methods

### 3.1. Sample

This study employed data from CILS, one of the largest longitudinal studies on the life course development of immigrant children in the United States. The study was conducted in Miami/Ft. Lauderdale and San Diego, two major immigrant-receiving regions in the United States [26]. The sample included US-born immigrant children with at least one immigrant parent and youths who had immigrated to the US at a young age. The first wave of CILS was conducted in 1992 with a school-based sampling frame and surveyed 5262 immigrant youth respondents ($\bar{X}$_age_ = 14) in the 8th and 9th grades of 49 public and private schools, representative of the population of immigrant youth children in the regions. The wave-2 survey was conducted in 1995–96, and 4288 respondents (81.5%) of the original sample were re-interviewed at the time of their high school graduation ($\bar{X}$_age_ = 17). The third wave survey was conducted in 2002. CILS retrieved 3613 respondents who were in their young adulthood ($\bar{X}$_age_ = 24), representing 68.9% and 84.3% of the sample in the wave-1 and 2 surveys, respectively. The current study used data across the three waves of CILS stretching a 10-year life course of immigrant children’s adaptation. The attribution rate of CILS is lower than other general longitudinal national surveys conducted in the US recently [13,55,58]. In this study, a sample of 3344 respondents who had given information related to the adaptation outcomes of college graduation, engagement in postgraduate study, and first and current occupational attainments in young adulthood were included. The major limitations of CILS are the sample of immigrant children only restricted to the two immigrant-receiving regions and lack non-Hispanic White youths as comparison. However, CILS has the strengths of incorporating a large panel sample of immigrant children from 77 nationalities and containing fruitful and useful individual, family, and school variables that pertain to their adaptation and development in young adulthood.

### 3.2. Measures

#### 3.2.1. Outcome Variables of Successful Adaptation

College graduation of immigrant children means the immigrant youth successfully obtained a 4-year undergraduate degree in higher education in the wave-3 survey of CILS, which is a dichotomous variable (1 = yes, 0 = no).

Engagement in postgraduate study connotes if the immigrant youth had obtained a master’s or professional/doctoral degree or currently enrolled in a graduate or professional degree of postgraduate schools in the wave-3 survey of CILS, which is a dichotomous variable (1 = yes, 0 = no).

First and current occupational attainments refer to the Treiman job prestige scores of the first and current occupations, in which the first occupation means the first job that the immigrant youth worked for when left schooling and the current occupation means the job she/he was doing currently at the time of wave-3 survey of CILS. The Treiman job prestige scores of the first and current occupations were provided by the principal investigators of CILS to indicate the consensual nature of the rating worthiness of a job based on its social prestige and admiration with a score ranging from 0 to 100 [26,59], higher scores meaning better occupational achievement.

#### 3.2.2. Structural Predictors of Segmented Assimilation Theory

Family composition, family SES, and incorporation mode are reported by the segmented assimilation theory as the most influential structural predictors of immigrant children’s successful adaptation [3,15,26]. In this study, family composition, family SES, and incorporation mode were included as both individual- and school-level variables to predict college graduation, engagement in postgraduate study, and first and current occupational attainments of immigrant children in young adulthood. Family composition is a dichotomous variable composited from two questions in the wave-1 survey of CILS by asking whether the immigrant youth was living with the biological father and mother, in which a two-parent-family status was assigned if the immigrant youth reported living with both biological father and mother (1 = two-parent family, 0 = otherwise). Family SES was measured by a unit-weighted standardized scale of parents’ education, home ownership, and occupational SES index, higher scores indicating better family SES status [30,33]. The incorporation mode is a dichotomous variable measuring if the immigrant youth received a negative reception or not due to their ethnicity (1 = negative, 0 = positive/neutral). Specifically, immigrant children of Haitian, Jamaican/West Indian, Mexican, or Nicaraguan origin were assigned to negative reception [15,26]. At the school level, family composition and incorporation mode are defined as the percentages of immigrant youths coming from two-parent households or receiving negative reception in the same school. School-level family SES is the average of individual-level family SES across immigrant youths in the same school.

#### 3.2.3. Structural Predictors of Conventional Assimilation Theory

Structural variables related to the conventional assimilation perspective include the generation of immigrant children, length of residence, English use, and interaction with native friends adopted from wave-1 data of CILS. For generation status, immigrant children are classified into 1.5 generation (immigrated with both parents to the US), 2 generation (born in the US with two immigrant parents), and 2.5 generation (born in the US with one immigrant parent and one native parent) [26,31]. Length of residence is a continuous variable (1 = less than five years, 2 = five to nine years, 3 = ten years or more, and 4 = all my life) [5]. English use is measured if the immigrant youth mainly used English for communication (1 = yes, 0 = no). Interaction with native friends is measured from a question asking the immigrant youth how many close friends were from abroad (1 = many or most, 2 = some, 3 = none), which is an ordinal variable.

#### 3.2.4. Mental Forces of Immigrant Children

The predictor of mental forces of immigrant children was constructed as a second-order latent factor by loading self-esteem and positive educational expectation convergently. In the wave-1 and 2 surveys of CILS, the well-validated 10-item Rosenberg Self-Esteem Scale was used to measure immigrant children’s self-concept [46]. Wave-1 and 2 Cronbach alpha coefficients of immigrant children’s self-esteem are α = 0.815 and 0.843, respectively. Educational expectation was measured by combing two questions asking immigrant children the highest education attainment they hoped to achieve and could be attained realistically: (1) “What is the highest level of education you would like to achieve?” and (2) “And realistically speaking, what is the highest level of education that you think you will get?”, which were measured on a 5-point scale (1 = less than high school, 2 = finish high school, 3 = finish some college, 4 = finish college, and 5 = finish a graduate degree). Since emphasizing academic success is the main duty of both general and immigrant youths in high school and will affect their future life trajectories [3,13,48], these two questions measured in the wave-1 and 2 surveys of CILS were combined as the educational expectation of immigrant youths, higher scores representing more positive educational expectation [3,4,26]. Wave-1 and -2 Cronbach alpha coefficients of immigrant children’s educational expectation were α = 0.805 and 0.815, respectively. The correlation coefficients of wave-1 and -2 self-esteem and wave-1 and -2 positive educational expectation were r = 0.445 and 0.509, ps < 0.001, and the correlation coefficients of wave-1 and -2 self-esteem with wave-1 and 2 educational expectations were from r = 0.179 to r = 0.261, ps < 0.001, proving their interrelationship for constructing a second-order latent factor.

To construct a second-order latent predictor of mental forces of immigrant youths, we first used a parceling strategy of “item-to-construct balance approach” to construct a first-order latent factor of immigrant children’s self-esteem for parsimony [60]. Due to the intercorrelations of the items, a principal component analysis of maximum likelihood and Promax rotation were conducted to examine the components and factor loadings of the measure [61], in which a two-factor solution with five items loading on each component emerged for the wave-1 self-esteem of immigrant children, Bartlett’s Test of Sphericity, X^2^ = 13,647.511, *p* < 0.001, indicating sampling and correlation adequacy for factoring. The factor loadings of the first and second components ranged from 0.526 to 0.818 and from 0.413 to 0.866, respectively. For wave-2 immigrant children’s self-esteem, the same two-factor solution emerged, in which sampling and correlation adequacy were confirmed, Bartlett’s X^2^ = 21,249.633, *p* < 0.001, and factor loadings of the first and second components ranged from 0.448 to 0.901 and from 0.428 to 0.887, respectively. Therefore, the items of the first and second components were parceled together to form a four-indicator first-order latent factor of immigrant children’s self-esteem. Moreover, the two items of immigrant children’s positive educational expectation in wave-1 and 2 surveys were loaded together to form the first-order latent factor. Then, the first-order latent factors of immigrant children’s self-esteem and educational expectation were loaded on a second-order latent factor to form mental forces of immigrant children. Based on a confirmatory factor analysis, the composite reliability of the immigrant children’s mental forces was *ρ*ϲ = 0.972, with model fit of comparative fit index (CFI) = 0.995, root mean square error of approximation (RMSEA) = 0.030, and standardized root mean-square residual (SRMR) = 0.014, supporting the convergence of self-esteem and positive educational expectation of immigrant children on a latent construct of mental forces.

#### 3.2.5. Individual-Level Control Variables

Individual-level control variables include gender, age, siblings, total mathematics and reading scores of standardized Stanford Achievement Tests (SAT10), and ethnic origin of immigrant children measured in wave-1 survey of CILS. Compared to their male counterparts, female immigrant children have greater academic success, called the “gender paradox” [6,13]. Older-aged immigrant children compared to their eligible-aged peers in school and immigrant children of more siblings have poorer adaptation, which are resulted from apportioned family resources and weakened parental care [3,30]. Standardized test scores have been reported more objectively indicative of children’s cognitive and learning abilities than teacher’s reported school grades [5,62]. Total mathematics and reading scores of standardized SAT10 were provided by the participating schools of CILS. In this study, gender is a dummy variable (female = 1, 0 = male), and age and siblings are count variables. As ethnic differences of immigrant children may shape their different adaptation outcomes [3,26,27], CILS surveyed immigrant children from 77 nationalities. This study classified them into eight ethnic groups according to their cultural and historical backgrounds [15,18]: Cuban, Mexican, Caribbean, Central and South American, Southeast Asian, Northeast/East Asian, Middle Eastern and African, and European. Research showed that Mexican and Caribbean immigrant children inclined to downward assimilation and their Northeast/East Asian counterparts did the opposite and performed much better; the remaining ethnic groups were in between [3,4,27]. However, recent empirical findings did not support these ethnic differences after controlling the family, assimilation, and cultural variables [5,13]. Due to the inconclusive nature of ethnic differences, we set immigrant children of Northeast/East Asian origin, e.g., Chinese, Korean, and Japanese, as reference due to their good academic and social performance and placed the remaining in comparison [19,27,33], thereby constituting seven ethnic dummy variables.

#### 3.2.6. School-Level Control Variables

School-level control variables include school type, minority school status, school location, and school academic achievement. School type is a dummy variable (1 = public school, 0 = private school), which corresponds to prior research reporting that students in private schools performed better academically than their peers in public schools [13,26,33]. Minority school status indicates if the school had 60% or more students with minority background (1 = minority school, 0 = otherwise), which needs to be controlled at school level, as schools mainly enrolling minority students showed poorer educational standards [15,33]. School location refers to whether the school is inner-city or not (1 = inner-city, 0 = otherwise), for which research reported that educational quality and performance of inner-city schools are poorer than those located in rural or suburban areas [3,31,33]. School academic achievement represents students’ total mathematics and reading scores of standardized SAT10 aggregated at the school level, which have been indicated as a more objective measurement of school-level educational performance than students’ GPA [32,33].

### 3.3. Modeling Procedures

Due to the clustered sampling structure of CILS data and our assumptions of immigrant children’s successful adaptation embedded in both individual and school levels, multilevel modeling is used in this study, and we employed a Bayesian analysis approach rather than a frequentist framework to take the advantage of enhancing external validity and accounting for uncertainties to avoid biased results [63,64]. Recent simulation studies have shown that the use of Bayesian framework as an estimation in advanced modeling procedures obtains more reliable parameter estimates and precludes an inflation of Type I errors, which are commonly found in maximum likelihood estimations (ML) applied in the frequentist framework [65,66]. The Bayesian analysis approach views data as fixed and parameters as random [64,66], in which parameter estimates are based on distributions rather than single-point values employed by frequentist procedures. Bayesian modeling involves the processing quantities of prior distribution, likelihood, and posterior distribution through Markov chain Monte Carlo (MCMC) estimation algorithms that are the techniques of sampling by probability distribution using Markov chains [67,68]. After a weighted integration of prior distribution and likelihood, posterior distribution can be formed by several thousand replications of possible parameter estimates. Therefore, the results of Bayesian modeling are not biased by multivariate non-normality, sampling imbalance, restricted variances, and conditional uncertainties [66,68]. The Bayesian theorem used in the Bayesian framework is:(1)$$P(\theta |y)=\frac{P(y|\theta )P\left(\theta \right)\;}{P\left(y\right)}$$
where P(θ|y) is the posterior distribution of the parameter concerned, P(y|θ) is the likelihood, $P\left(\theta \right)$ is the prior distribution, and $P\left(y\right)$ is the probability of the data. Accordingly, a unifying form of multilevel modeling in Bayesian framework is written as:(2)$$\eta =\chi \beta +\overset{L}{\underset{l=1}{\sum }}Z_{1}U_{1}+\varepsilon =\eta +\varepsilon ,$$
(3)$$\varepsilon \sim N\left(0,\;\sigma^{2}I\right),$$
where *η* is the *n* × 1 vector of observations, *β* is the *p * × 1 vector of unknown fixed coefficients, *Z*_1_ and $U_{1}$ represent Gaussian random effect vectors at the *Ɩ*th cluster level [68,69], which are applicable to generalized linear multilevel modeling by incorporating exponential family distributions and conditional expectations through the following link function:(4)$$E\left(y\right)=g\left(\eta \right),$$
in which the fitting of parameter estimates is conducted by simulation of MCMC sampling techniques [67,70], and the potential scale reduction factor (*PSR*) is used to determine the convergence. *PSR* has the form:(5)$$PSR=\sqrt{\frac{I_{p}+S_{p}}{S_{p}}}\;,$$
where $I_{p}$ is individual-level chain variance for parameter *p*, and $S_{p}$ is the school-level chain variance for parameter *p.* If *PSR* ≤ 1.05, the MCMC algorithm process is stopped, and modeling convergence is successfully attained.

## 4. Results

Of the 3344 immigrant children, 54.1% (*n* = 1809) were female. The mean age of the sample in wave 1 was 14.23 years (Table 2). Many immigrant children were 1.5 generation (46.9%, *n* = 1567) and 2 generation (41.3%, *n* = 1380), and 2.5-generation made up only 11.9% (*n* = 397). They generally have lived in the US between 5 and 9 years (mean = 1.87). A quarter came from non-two-parent families (24.2%, *n* = 810), and the mean score of family SES calculated from their parents by the Duncan Socioeconomic Index (SEI) was 34.25, which was much lower than the US general citizens compared by the data of the 1994 General Social Survey (ranging from 44.2 to 48.7). Nearly one-third of immigrant children had a negative reception (27.1%). Most were from Southeast Asia and Cuba (28.9% and 24.9%), followed by Central/South America, Mexico, and the Caribbean (16.6%, 12.4%, and 10.9%). The fewest came from Northeast/East Asia, Middle East/Africa, and Europe (2.3%, 1.9%, and 1.9%).

Multilevel unconditional models confirmed certain variances of immigrant children’s adaptation outcomes existing at both the individual and school levels (Table 3). Intraclass correlations (ICCs) ranged from 0.056 to 0.161, and the significance of Wald Z values indicates conducting a multilevel modeling validly. Table 4 shows the multilevel modeling effects of social forces and pertinent individual and ethnic covariates on the four adaptation outcomes of immigrant children, in which the individual-level predictors of two-parent families, family SES, and negative reception were significantly and robustly predictive of immigrant children’s college graduation, engagement in postgraduate study, and first and current occupational attainments. Specifically, immigrant children raised in two-parent families had the increased odds of successful college graduation and engagement in postgraduate study by 9.8% and 12%, respectively; they also significantly and marginally significantly performed better in their first occupation, β = 0.063, *p* < 0.001, and current occupation, β = 0.033, *p* = 0.062. In addition, a unit increase in family SES was significantly predictive of college graduation and engagement in postgraduate study by the increased odds of 27.3% and 25.8%, respectively. Immigrant children of higher family SES had better performance in their first and current occupations, βs = 0.110 and 0.125, ps < 0.001. The importance of family SES also presents at the school level. Immigrant children attending schools of higher aggregated family SES by one unit significantly contributed to their successful college graduation and engagement in postgraduate study by the increased odds of 48.5% and 11.5%, ps < 0.001, and higher aggregated family SES at school level also significantly predicted their better first and current occupational attainments, βs = 0.425 and 0.562, ps < 0.05 and 0.01. Immigrant children of negative reception, however, significantly performed poorer in their first and current occupations, βs = −0.052 and −0.040, ps < 0.01 and 0.05. Nevertheless, the above multilevel models explain only limited variances of immigrant children’s adaptation outcomes of college graduation, engagement in postgraduate study, and first occupation and current occupation attainments, with the variances explained ranging from 5.2% to 10.8% at the individual level and the variances explained ranging from 62.8% to 72.7% at the school level. In addition, the total variances explained by these multilevel models were 10.3% for college graduation, 14.3% for engagement in postgraduate study, 10.6% for first occupational attainment, and 9.9% for current occupational attainment.

We conducted another multilevel modeling by incorporating immigrant children’s mental forces as a latent predictor of their adaptation outcomes in young adulthood. Table 5 shows the factor loadings of first-order latent factors of immigrant children’s self-esteem and educational expectation and the second-order latent predictor of mental forces across multilevel models of college graduation, engagement in postgraduate study, and first and current occupational attainments. The results support that all first-order parcels, indicators, and second-order factors were significantly loaded on their respective latent constructs. Specifically, the factor loading coefficients of first-order factors across the four adaptation outcomes of college graduation, engagement in postgraduate study, and first and current occupation attainments are from λ = 0.509 to 0.834, ps < 0.001, and the factor loading coefficients of the second-order construct across the four adaptation outcomes are from λ = 0.440 to 0.894, ps < 0.001, which are all beyond the strict threshold of suggested λ ≥ 0.400. 

Table 6 presents the effects of mental forces of immigrant children on their adaptation outcomes of college graduation, engagement in postgraduate study, first occupation and current occupation attainments. A unit increase in mental forces of immigrant children was significantly related to the higher odds of their college graduation by more than 11 times and engagement in postgraduate study by more than 18 times. Moreover, mental forces of immigrant children significantly and substantially predicted their better first and current occupational attainments, βs = 0.293 and 0.297, ps < 0.001. Additionally, immigrant children in two-parent families significantly had the higher odds of college graduation by 8.8% and engagement in postgraduate study by 11.5%; they also significantly performed better in their first occupation, β = 0.050, *p* < 0.01. Furthermore, a unit increase in family SES significantly contributed to the higher odds of college graduation by 22.2% and engagement in postgraduate study by 21.4%. Family SES also significantly and positively predicted better first and current occupational attainments, βs = 0.078 and 0.092, ps < 0.01. In contrast, immigrant children of negative reception performed poorer in their first and current occupations, βs = −0.044 and −0.032, ps < 0.05 and 0.1. School-level family SES had significant and strong effects on educational and occupational success of immigrant children: a unit increase in school-level family SES contributed to the higher odds of college graduation by 43.9% and engagement in postgraduate study by nearly three times. Higher school-level family SES also significantly predicted better first and current occupational attainments of immigrant children in young adulthood, βs = 0.463 and 0.639, ps < 0.01. Combining both the effects of social and mental forces in multilevel modeling improved individual-level and total variances explained for the adaptation outcomes of immigrant children. Specifically, individual-level variances explained for the adaptation outcomes of immigrant children were increased to 12.1% and 37.1%, and total variances explained were also increased to 18.3% for college graduation, 23% for engagement in postgraduate study, 18.7% for first occupational attainment, and 18.2% for current occupational attainment.

To compare whether the effects of mental forces of immigrant youths were significantly stronger than the structural effects of family composition, family SES, and incorporation mode on their adaptation outcomes, we performed Wald tests of the parameter constraints by setting equivalence of the aforementioned effects on immigrant children’s college graduation, engagement in postgraduate study, and first and current occupational attainments; that is, β_mental forces_ = β_two-parent family_, β_family SES_, and β_negative inception_. Wald X^2^ values showed that the effects of mental forces were significantly stronger than those of two-parent families, family SES and negative reception on the four adaptation outcomes of immigrant children (Table 7). Comparatively, the effects of mental forces were 2.863–3.756 times stronger than those of family SES on immigrant children’s adaptation outcomes. In addition, the effects of mental forces were 5.860–6.722 times stronger than the effects of two-parent families on immigrant children’s adaptation outcomes. Lastly, mental forces had a significant effect of 6.659 times stronger than that of negative reception on immigrant children’s first occupational attainment.

To investigate whether mental forces of immigrant children during adolescence had the varying effects on their adaptation outcomes conditioned by the contexts of family composition, family SES, and incorporation mode, immigrant children were divided into two comparative groups by family composition (two-parent vs. non-two-parent), family SES (high vs. low), and reception mode (negative vs. positive/neutral). The same multilevel modeling procedures were conducted again. Table 8 shows that the effects of mental forces on college graduation were significantly stronger for immigrant children in the unfavorable contexts of non-two-parent families, low family SES, and negative reception than their counterparts in the favorable contexts of two-parent families, high family SES, and positive/neutral reception. Similarly, z-tests supported certain evidence of the stronger effects of mental forces of immigrant children on their engagement in postgraduate study in the unfavorable contexts of non-two-parent families, low family SES, and negative reception compared to their counterparts in the favorable contexts of two-parent families, high family SES, and positive/neutral reception. Noteworthy is the highly significant different effect of immigrant children’s mental forces in the context of negative reception compared to their counterparts in the context of positive/neutral reception on engagement in postgraduate study, β_difference_ = 0.295, Z = 4.214, *p* < 0.001. Furthermore, as hypothesized, the significant effects of mental forces of immigrant children on their first occupational attainment were comparable across the contrasting contexts of family composition, family SES, and incorporation mode. Furthermore, there was no difference in the significant effects of immigrant children’s mental forces on their current occupational attainment across the different contexts of family composition, family SES, and incorporation mode. Moreover, immigrant children of low family SES had a marginally significantly stronger effect of mental forces on their current occupational attainment as compared to their counterparts of high family SES, β = 0.251 vs. 0.318, ps < 0 0.001, β_difference_ = −0.067, *p* < 0.1. Nevertheless, no significant different effects of immigrant children’s mental forces on their first and current occupational attainments in the different contexts of incorporation mode (positive/neutral reception vs. negative reception) were found.

## 5. Discussion

The present study is the first attempt of immigration research to consider and examine both the effects of social and mental forces in adolescence of immigrant children at individual and school levels on their successful adaptation in young adulthood by concurrently investigating multiple adaptation outcomes of college graduation, engagement in postgraduate study, and first occupation and current occupation attainments. An overall picture from the findings is that both the effects of social and mental forces affect successful adaptation of immigrant children, but the latter appears to be more robust but also conditioned by the former—that is, the social forces suggested by segmented assimilation theory. In the first place, results of multilevel modeling supported that family composition, family SES, and incorporation mode stressed by the segmented assimilation theory more significantly and consistently contributed to immigrant children’s adaptation outcomes than generational status, length of residence, English use, and interaction with native friends upheld by the conventional assimilation theory. In fact, family composition, family SES, and incorporation mode can generate most proximal and intimate socialization experiences to immigrant children for their cognitive, personality, and behavioral development relating to educational and occupational success [26,28,31]. Immigrant children in two-parent families, higher family SES, and positive/neutral reception conditions denote having better family support and resources, parental education, inductive parenting, and social capital, which are important for their successful adaptation in adulthood, especially critical for children living in a new, strange, and culturally different host society [18,22,35]. In contrast, immigrant children of non-two-parent families, lower family SES, and negative reception background indicate insufficient resources, role models, parental support, and interpersonal network, which constitute unfavorable socialization experiences for their positive growth [11,22,27], hence attributing to their maladaptation and “significantly greater probability of downward assimilation relative to other groups” [71] (p. 21). However, the effects of negative reception on the two educational outcomes were not significant in multilevel modeling, reflecting the importance of positive family socialization and resources, as well as procedural equality of higher education in Western societies to determine educational success of immigrant children rather than their incorporation status [3,56]. Manifestly, family SES has been empirically confirmed as a very important family component to transmit parental human and social capitals, such as capability, education, cognition, and occupational status, to their offspring generations by the form of steering their life trajectories and adaptation [22,28,35]. Relevantly, the current study found that the significant effects of family SES on immigrant children’s adaptation outcomes keep robustly at school level, which may reflect the invisible but powerful resource-driven social selectivity process of Western capitalist societies at work under the aegis of so-called political democracy and equalitarianism [26,72].

One of the main purposes of this study was to examine and compare the effects of mental forces of immigrant children to the structural impacts of family composition, family SES, and incorporation mode proposed by the segmented assimilation theory. The results of the multilevel modeling showed that the effects of immigrant children’s mental forces measured in adolescence significantly and substantially predicted their educational and occupational outcomes in young adulthood. Evidently, the effects of mental forces on immigrant children’s successful adaptation are significantly stronger than the influences of family composition, family SES, and incorporation mode emphasized by segmented assimilation theory. For easier interpretation of the contrasts between the effects of mental forces and those of segmented assimilation theory on immigrant children’s adaptation outcomes, we present their differences with Bayesian posterior parameter trace plots. Figure A1 displays the different effects of mental forces and those of segmented assimilation theory on immigrant children’s college graduation run by Bayesian estimation with simulation integration of MCMC sampling by setting Fbiterations = 5000 (see Appendix A). The latent effects of mental forces were soon converged after starting iterations of repeated random sampling and were explicitly stronger than those of family composition, family SES, and incorporation mode throughout the random walk process of the MCMC algorithm. Figure A2, Figure A3 and Figure A4 are the Bayesian posterior parameter trace plots presenting the converging results on immigrant children’s engagement in postgraduate study, and first and current occupational attainments in young adulthood, and similar converging patterns of different effects of mental forces and those of segmented assimilation theory appeared again (see Appendix A).

The results of the current study reveal that mental resources and intrapersonal strengths of immigrant children play an integral and critical role in relation to their later successful adaptation, demonstrating the significance of human will and determination in shaping later life trajectories, which even account for the structural effects proposed by the conventional assimilation and segmented assimilation theories. Portes and Fernandez-Kelly [71] claimed the educational and occupational achievements of disadvantaged immigrant children as a “no margin for error” process directed by their parents through stressing strict socialization and preservation of ethnic culture and values to keep their children’s perseverance for positive development. Undeniably, the above assertion by Portes and Fernandez-Kelly [71] belongs to parental concerns of cultivating their immigrant children’s unfailing determination, efforts, and confidence to overcome contextual obstacles and difficulties for their future success and upward mobility in “the American hierarchies of status and wealth” (p. 13). Pertinently, resilient mental forces of immigrant children are much stronger than the structural effects suggested by the segmented association theory in relation to their successful adaptation in young adulthood. This corresponds to Portes and Fernandez-Kelly [71] emphasizing the mental assets acting as “a cultural reference point on which to ground their [immigrant children’s] sense of self and their personal dignity…this reference point is also an important component of success stories (p. 24)”. In fact, both the individual-level and total variances explained in multilevel modeling indicate that immigrant children’s adaptation outcomes increased conspicuously after including mental forces of immigrant children, corroborating its rigorous function to sway the prospective achievements of immigrant children dynamically and temporally. Therefore, immigration research scholars should consider the interplay of intrapersonal traits and contextual influences in an interactive way when investigating immigrant children’s adaptation.

Nevertheless, it is noted that the effects of mental forces on immigrant children’s adaptation outcomes are significantly conditioned by the structural contexts of family composition, family SES, and incorporation mode related to the segmented assimilation theory (Table 8). This study found that the effects of mental forces on the two educational outcomes were significantly stronger for immigrant children in non-two-parent families, lower family SES, and negative reception conditions than did their counterparts of two-parent families, higher family SES, and positive/neutral reception. This is congruent with our assumption of the context-driven resilient response thesis of immigrant children’s mental forces, in which their intrapersonal determination, confidence, and strengths are more salient and helpful for their successful adaptation if the adaptation outcomes are more controllable and attainable by personal efforts and resolution, such as college graduation and engagement in postgraduate study. In other words, if immigrant children think that they can change their disadvantaged life situations by achieving certain manageable adaptation outcomes [49,53,73], the effects of mental forces would function more potently. Validly, the above-mentioned empirical evidence of context-driven resilient response, is more visible in the effects of mental forces on immigrant children’s engagement in postgraduate study in the context of incorporation mode (negative vs. positive/neutral), because obtaining a higher degree can help negatively received immigrant children to equalize their humble social positions with their better-off counterparts. Future research should pay attention to the nature of adaptation outcomes and structural contexts in regulating immigrant children’s mental forces for their future success.

Furthermore, the conditional effects of mental forces of immigrant children on the two occupational outcomes were insignificant across the different contexts of family composition, family SES, and incorporation mode proposed by the segmented assimilation theory, which cast the possibility of nullifying the effects of uncontrollable extraneous factors, such as job market selection, family resources, and interpersonal network, in confounding the effects of immigrant children’s mental forces on those less personally controllable adaptation outcomes [44,50,74]. Nevertheless, the significant effects of mental forces of immigrant children on their first and current occupational attainments still persist across the favorable and unfavorable contexts of family composition (two-parent vs. non-two-parent), family SES (high vs. low), and incorporation mode (positive/neutral vs. negative), revealing the consistent importance of immigrant children’s mental resources and determination in relation to their future achievements. However, the current study only found little evidence to support hypothesis 8 in expectation that, after accumulating certain social and working experiences, the stronger significant effects of immigrant children’s mental forces in the unfavorable contexts of non-two-parent families and lower family SES would resume. One explanation is that immigrant children at wave-3 CILS were, on average, 24 years old, and they still need to develop more human and social capital and work experiences to let their mental forces function effectively to change their disadvantaged career path derived from their unfavorable family and social contexts. Hence, a longer panel design in tracing transitions of immigrant children’s career development in different contexts affected by the effects of immigrant children’s mental forces is suggested to examine the tenability of the context-driven resilient response thesis for occupational outcomes in the future.

Although the current study shows the importance of social forces proposed by the segmented assimilation theory and mental forces of immigrant children during adolescence in contribution to their successful adaptation in young adulthood, the main drawbacks of this study are that the immigrant children were mainly drawn from Miami/Ft. Lauderdale and San Diego, and CILS only contains available data of immigrant children’s self-esteem and future educational orientation and lacks US-born White counterparts as reference. If we can research the effects of mental and social forces and their interplay with data based on more rigorous sampling and research designs to include both cross-country and representative samples of immigrant and native children and also prolong the transitional changes of immigrant children, new insights regarding immigrant children’s long-term adaptation and life trajectories can be revealed. In addition, the data of CILS were collected in the 1990s and early 2000s, during which, due to the dominant views of conventional assimilation and segmented assimilation theories and endorsement of economic liberalism, other information such as intervention programs of inclusive education and multicultural competence services for equalizing development of immigrant children were not available. For this, immigration researchers should incorporate data of intervention services or policy supports, if available, as moderators or mediators when investigating the effects of social and mental forces on the long-term adaptation of immigrant children. In sum, immigration research scholars in the future are suggested to include not only the effects of social and mental forces, as well as their interactive influences on adaptation of immigrant children, but also need to adopt a more culturally and temporally diverse research design for clearly examining the complicated processes of immigrant children’s adaptation and life trajectories.

## 6. Conclusions

It is undeniable that existing available immigration studies preponderantly stress the effects of structural factors on adaptation of immigrant children but overlook the importance of school contextual influences and impacts of immigrant children’s intrapersonal traits and mental resources in relation to their long-term development. In the current study, we found that the developmental trajectories of immigrant children are simultaneously affected by the effects of social and mental forces at individual and school levels, in which mental forces of immigrant children are evidenced more profoundly influential on their adaptation outcomes of college graduation, engagement in postgraduate study, and first occupation and current occupation attainments in young adulthood than those structural effects of social forces related to segmented assimilation theory. Nevertheless, the effects of mental forces of immigrant children on their adaptation are also corroborated conditional by the different contexts of social forces suggested by segmented assimilation theory, which support our proposed thesis of the context-driven response. In sum, it is not only researchers and scholars should consider these complicated and dynamic adaptation processes of immigrant children, but also policymakers and educators should account for the proximal environmental conditions and mental outlook of immigrant children interactively when designing interventions and providing supports.

## Figures and Tables

**Table 1 ijerph-19-06433-t001:** Anticipated Conditional Effects of Mental Forces of Immigrant Children on Adaptation Outcomes across the Different Contexts of Family Composition, Family SES, and Incorporation Model in Young Adulthood.

	Family Composition ^a^		Family SES ^b^		Incorporation Mode ^c^	
	Two-Parent	Non-Two-Parent	High	Low	Positive/Neutral	Negative
College Graduation	L ^d^	H ^d^	L ^d^	H	L	H
Engagement in Postgraduate Study	L	H	L	H	L	H
First Occupational Attainment	N ^d^	N	N	N	N	N
Current Occupational Attainment	L	H	L	H	N	N

Note: ^a^ Two-parent families represents immigrant children living with two biological parents in the same household, and non-two-parent families means otherwise. ^b^ Family SES refers to the family socioeconomic status measured by the unit-weighted standardized scores of parents’ education, home ownership, and occupational socioeconomic index, in which higher than the mean of the study population indicates high family SES and otherwise is assigned to low family SES. ^c^ Immigrant children of negative reception include those of Haitian, Jamaican/West Indian, Mexican, or Nicaraguan origin and otherwise belong to positive/neutral reception. ^d^ L connotes the effects of mental forces on immigrant children’s adaptation outcomes are less pronounced, H connotes the effects of mental forces on immigrant children’s adaptation outcomes are more salient, and N means there are no differences between the effects of mental forces on immigrant children’s adaptation outcomes across contexts.//

**Table 2 ijerph-19-06433-t002:** Demographics of Immigrant Children of CILS, *n* = 3344.

	Mean/Proportion	SD	Range
Gender			
Female	0.541		0, 1
Male	0.459		0, 1
Age	14.234	0.863	12–18
Siblings	1.797	1.463	0–8
US Born			
Yes	0.561		0, 1
No	0.469		0, 1
Generation			
1.5 generation	0.469		0, 1
2 Generation	0.413		0, 1
2.5 Generation	0.119		0, 1
Residence	1.87	0.953	1–4
Family Composition			
Two-parent Family	0.758		0, 1
Non-two-parent families	0.242		0, 1
Family SES	34.253	13.043	13–88
Incorporation Mode			
Positive/Neutral	0.729		0, 1
Negative	0.271		0, 1
Ethnic Background			
Cuban	0.249		0, 1
Mexican	0.124		0, 1
Caribbean	0.109		0, 1
Central/South American	0.166		0, 1
Southeast Asian	0.289		0, 1
Northeast/East Asian	0.023		0, 1
Middle East/African	0.019		0, 1
European	0.019		0, 1

Note: Mean is for continuous variables, and proportion is for dummy or dichotomous variables. Siblings mean the number of siblings in the same household. two-parent families refer to immigrant children who live with both biological father and mother, and non-two-parent families mean otherwise. For the generation of immigrant children, 1.5 generation indicates immigrant children who immigrated with both parents to the US, 2 generation means those born in the US with two immigrant parents, and 2.5 generation refers to immigrant children born in the US with one immigrant parent and one native parent. Family SES is a unit-weighted standardized scale of parents’ education, home ownership, and occupational socioeconomic index.

**Table 3 ijerph-19-06433-t003:** Unconditional Modeling Immigrant Children’s Adaptation Outcomes in the Individual and School Levels.

Adaptation Outcomes		Estimate	SE	Wald Z	ICC
College Graduation	Individual Level	0.216	0.005	42.843 ***	0.056
	School Level	0.013	0.004	3.014 **	
Postgraduate Study	Individual Level	0.114	0.015	7.591 ***	0.161
	School Level	0.022	0.007	3.041 ***	
First Occupational Attainment	Individual Level	97.910	2.714	36.070 ***	0.082
	School Level	8.756	2.031	4.312 ***	
Current Occupational Attainment	Individual Level	107.589	3.743	28.742 ***	0.073
	School Level	8.581	1.988	4.315 ***	

Note: ICC = Intraclass correlation. + *p* < 0.1; * *p* < 0.05; ** *p* < 0.01; *** *p* < 0.001.

**Table 4 ijerph-19-06433-t004:** Multilevel Modeling Effects of Social Forces on Adaptation Outcomes of Immigrant Children in Young Adulthood.

	Outcomes	College Graduation	Postgraduate Study	First Occupational Attainment	Current Occupational Attainment
Predictors		OR	OR	β	β
Individual Level					
Female		1.136 ***	1.112 ***	0.139 ***	0.127 ***
Age		0.954 ^+^	0.932 ^+^	−0.031	−0.008
Siblings		0.972	0.939 ^+^	−0.021	−0.035 ^+^
Stanford Math Scores		0.976	1.001	0.031	0.070 ***
Stanford Reading Scores		1.016	1.000	−0.029	−0.048 *
1.5 Generation		1.254 **	1.099	0.091 *	0.112 **
2 Generation		1.254 **	1.136	0.058 ^+^	0.099 **
Residence		1.031	1.016	−0.041	0.001
English Only		1.019	1.038	−0.017	0.003
Native Friendship		0.976	0.974	−0.021	−0.026
Two-parent Family		1.098 ***	1.120 ***	0.063 ***	0.033 ^+^
Family SES		1.273 ***	1.258 ***	0.110 ***	0.125 ***
Negative Reception		0.955	1.008	−0.052 **	−0.040 *
Cuban		1.066	1.132	0.097	0.014
Mexican		1.097	1.302	0.099 *	0.032
Caribbean		1.110	1.180	0.072	0.044
Central and South American		0.986	1.042	0.087 ^+^	0.019
Southeast Asian		0.1069	1.149	0.075	0.019
Middle East/African		1.392 ^+^	0.778	0.016	−0.003
European		0.962	0.857	0.004	−0.017
School Level					
Public School		0.982	1.071	0.018	−0.118
Minority School		1.038	1.225	0.394 ^+^	0.236
Inner-city School		0.941	1.147	−0.096	−0.026
Aggregated Stanford Math Scores		1.059	1.215	0.007	0.005
Aggregated Stanford Reading Scores		0.941	0.822 ^+^	−0.082	−0.053
Two-parent Family Proportion		1.149	0.617	0.078	−0.054
Aggregated Family SES		1.485 ***	1.115 ***	0.425 *	0.562 **
Negative Reception Proportion		0.968	0.832	−0.113	−0.106
Intercept τ_00_		2.153 ***	2.090 ***	13.490 ***	16.083 ***
Residual σ^2^		0.273 ***	0.372 ***	0.300 ***	0.278 ***
$R_{within}^{2}$		0.104	0.108	0.056	0.052
$R_{between}^{2}$		0.727	0.628	0.700	0.722
$R_{total}^{2}$		0.103	0.143	0.106	0.099

+ *p* < 0.1; * *p* < 0.05; ** *p* < 0.01; *** *p* < 0.001.

**Table 5 ijerph-19-06433-t005:** Structures and Loadings of the Second-order Latent Factor of Resilient Mental Forces in Immigrant Children across Multilevel Models.

		College Graduation		Postgraduate Study		First Occupation		Current Occupation	
		λ	Z	λ	Z	λ	Z	λ	Z
First-order Loadings									
1.	Wave-1 Parcel 1 -> Self-esteem	0.510	28.478 ***	0.508	27.848 ***	0.509	28.164 ***	0.509	27.824 ***
2.	Wave-1 Parcel 2 -> Self-esteem	0.533	28.645 ***	0.532	28.481 ***	0.532	28.761 ***	0.532	28.712 ***
3.	Wave-2 Parcel 1 -> Self-esteem	0.788	48.653 ***	0.788	48.213 ***	0.789	47.531 ***	0.789	47.774 ***
4.	Wave-2 Parcel 2 -> Self-esteem	0.723	44.440 ***	0.723	43.976 ***	0.722	45.021 ***	0.722	44.775 ***
5.	Wave-1 Indicator 1 -> Future Orientation	0.611	23.247 ***	0.615	22.024 ***	0.615	22.021 ***	0.613	22.047 ***
6.	Wave-1 Indicator 2 -> Future Orientation	0.620	22.301 ***	0.624	21.306 ***	0.624	21.346 ***	0.622	21.340 ***
7.	Wave-2 Indicator 1 -> Future Orientation	0.768	44.191 ***	0.766	41.971 ***	0.767	41.536 ***	0.768	41.765 ***
8.	Wave-2 Indicator 2 -> Future Orientation	0.834	49.059 ***	0.831	46.780 ***	0.831	45.301 ***	0.831	45.462 ***
Second-order Loadings									
9.	Self-esteem-> Resilient Mental Forces	0.463	18.645 ***	0.440	13.065 ***	0.572	15.841 ***	0.557	17.882 ***
10.	Future Orientation-> Resilient Mental Forces	0.894	21.315 ***	0.937	14.386 ***	0.720	14.524 ***	0.740	16.515 ***

Note. λ = Loading Coefficients; Z = Z-scores; + *p* < 0.1; * *p* < 0.05; ** *p* 0.01; < *** *p* < 0.001.

**Table 6 ijerph-19-06433-t006:** Multilevel Modeling Effects of Social and Mental Forces on Adaptation Outcomes of Immigrant Children in Young Adulthood.

	Outcomes	College Graduation	Postgraduate Study	First Occupational Attainment	Current Occupational Attainment
Predictors		OR	OR	β	β
Individual Level					
Female		1.098 ***	1.073 *	0.116 ***	0.103 ***
Age		0.992	0.966	−0.006	0.018
Siblings		0.997	0.972	−0.011	−0.025
Stanford Math Scores		1.000	0.984	0.019	0.060 **
Stanford Reading Scores		0.965	1.012	−0.022	−0.040 ^+^
1.5 Generation		1.212 ^+^	0.964	0.063 *	0.096 **
2 Generation		1.216 *	1.047	0.041	0.088 **
Residence		1.024	1.033	−0.037	0.001
English Only		1.015	1.038	−0.022	−0.002
Native Friendship		0.994	0.994	−0.011	−0.016
Two-parent Family		1.088 ***	1.115 ***	0.050 **	0.020
Family SES		1.222 ***	1.214 ***	0.078 ***	0.092 ***
Negative Reception		0.965	1.014	−0.044 *	−0.033 ^+^
Cuban		1.073	0.850	0.040	−0.009
Mexican		1.118	1.006	0.051	0.012
Caribbean		1.165	0.910	0.032	0.027
Central and South American		0.984	0.779	0.035	−0.003
Southeast Asian		1.066	0.873	0.015	−0.007
Middle East/African		1.414	0.574	−0.002	−0.011
European		0.988	0.651	−0.009	−0.020
Latent Mental Forces		11.290 ***	18.119 ***	0.293 ***	0.297 ***
School Level					
Public School		0.985	1.080	0.052	−0.121
Minority School		0.995	1.206	0.389 ^+^	0.197
Inner-city School		1.002	1.205	−0.007	0.096
Aggregated Stanford Math Scores		1.036	1.192	−0.134	−0.161
Aggregated Stanford Reading Scores		0.938	0.829	0.003	0.038
Two-parent Family Proportion		1.124	0.549	0.118	−0.054
Aggregated Family SES		1.439 **	2.959 ***	0.463 **	0.639 **
Negative Reception Proportion		0.967	0.770	−0.126	−0.106
Intercept τ_00_		2.731 ***	1.856 ***	16.872 ***	20.373 ***
Residual σ^2^		0.258 ***	0.387 ***	.245 ***	0.223 ***
$R_{within}^{2}$		0.293	0.371	.121	0.122
$R_{between}^{2}$		0.742	0.613	.755	0.777
$R_{total}^{2}$		0.186	0.230	.187	0.182

+ *p* < 0.1; * *p* < 0.05; ** *p* < 0.01; *** *p* < 0.001.

**Table 7 ijerph-19-06433-t007:** Comparing Standardized Effects of Mental Forces and Social Forces by Parameter Constraints from Multilevel Modeling.

	Predictors	Mental Forces ^a^		Two-Parent Family ^b^		Family SES ^c^		Negative Reception ^d^		Contrast	Wald X^2^ (df)
Outcomes		β	SE	β	SE	β	SE	β	SE		
College Graduation		0.484 ***	0.033	0.072 ***	0.022	0.169 ***	0.023	−0.030	0.025	a > b, c, d	82.261 (3) ***
Postgraduate Study		0.556 ***	0.042	0.086 ***	0.027	0.154 ***	0.027	0.011	0.032	a > b, c, d	78.306 (3) ***
First Occupation		0.293 ***	0.025	0.050 **	0.017	0.078 ***	0.019	−0.044 *	0.020	a > b, c, d	135.048 (3) ***
Current Occupation		0.297 ***	0.025	0.020	0.017	0.092 ***	0.019	−0.033 ^+^	0.020	a > b, c, d	110.166 (3) ***

Note: Odds ratios of college graduation and engagement in postgraduate study cannot be used directly to compare with standardized betas of two-parent families, family SES, and negative reception. Hence, standardized effects of mental forces were used to conduct Wald tests of parameter constraints by setting β_mental forces_ = β_two-parent family_, β_family SES_, and β_negative inception_ in model tests, and the same multilevel modeling of regressing both the effects of mental forces and social forces on immigrant children’s adaptation outcomes were performed. + *p* < 0.1; * *p* < 0.05; ** *p* < 0.01; *** *p* < 0.001.

**Table 8 ijerph-19-06433-t008:** Comparing Standardized Effects of Mental Forces on Immigrant Children’s Adaptation Outcomes across the Contexts of Family Composition, Family SES and Incorporation Mode from Multilevel Modeling.

Family Composition	Two-Parent		Non-Two-Parent		Difference in Beta (β_difference_)	Z-Value
	β_1_	SE	β_2_	SE	(β_1_ − β_2_)	
College Graduation	0.452 ***	0.038	0.597 ***	0.064	−0.145	−2.071 *
Postgraduate Study	0.536 ***	0.050	0.662 ***	0.080	−0.126	−1.415 ^+^
First Occupation	0.288 ***	0.030	0.272 ***	0.048	0.016	0.363
Current Occupation	0.290 ***	0.029	0.302 ***	0.048	−0.012	−0.268
**Family SES**	**Higher**		**Lower**		**Difference in Beta** **(β_difference_)**	
	**β_1_**	**SE**	**β_2_**	**SE**	**(β_1_ − β_2_)**	
College Graduation	0.400 ***	0.053	0.509 ***	0.050	−0.109	−1.730 *
Postgraduate Study	0.495 ***	0.063	0.629 ***	0.079	−0.134	−1.412 ^+^
First Occupation	0.270 ***	0.036	0.291 ***	0.035	−0.0021	−0.0477
Current Occupation	0.251 ***	0.035	0.318 ***	0.034	−0.067	−1.522 ^+^
**Incorporation Mode**	**Positive/Neutral**		**Negative**		**Difference in Beta** **(β_difference_)**	
	**β_1_**	**SE**	**β_2_**	**SE**	**(β_1_ − β_2_)**	
College Graduation	0.439 ***	0.042	0.546 ***	0.046	−0.107	−1.953 *
Postgraduate Study	0.524 ***	0.043	0.819 ***	0.067	−0.295	−4.214 ***
First Occupation	0.278 ***	0.031	0.272 ***	0.051	−0.006	0.095
Current Occupation	0.299 ***	0.031	0.267 ***	0.050	−0.032	0.727

+ *p* < 0.1; * *p* < 0.05; ** *p* < 0.01; *** *p* < 0.001.

## Data Availability

The data of CILS is available at The Center for Migration and Development in Princeton University, please visit https://cmd.princeton.edu/publications/data-archives/cils.

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
