# Peer review of "Where There’s a Will, There’s a Way? Social and Mental Forces of Successful Adaptation of Immigrant Children in Young Adulthood"

_ijerph, 2022, doi:10.3390/ijerph19116433_

Round 1

Reviewer 1 Report

This paper reports a thorough statistical analysis of a data set accessed from a US national sample: CILS with analyses of 3 waves of data scanning 10 years for 3344 young adults with immigrant backgrounds in selected locations. The analyses have been carefully constructed and reported. The authors have been careful to embed their statistical analyses in relevant theoretical work and empirical studies. It deserves publication, and I have a few comments that may make its dense and sometimes over-enthusiastic and turgid reporting a little more accessible for a diverse, international readership.

  1. Length and presentation. The paper is long and dense in its presentation. I suggest that:
  2. Some of the material be taken out of the main ms and be either attached as ‘Auxiliary material’ or be referred to in an end-note listing auxiliary material that the authors will make available on request. Either alternative is acceptable.
  3. Omit Table 1. It is a table of predictors and is difficult to read, and has too many notations that do not make sense. Place it in auxiliary material (1a).
  4. Omit the 4 Figures in the Discussion Section Place it in auxiliary material (1a).
  5. Is it possible to cut down the number of references, on their salience to the point being made and their recency.

  1. There are 9. They are stated clearly at the end of individual sections, but they are not specifically mentioned in the relevant Results Section reporting. I think the Heading 3 (top of p.3) ‘Conventional assimilation….” Is a little misleading and not very helpful. It could be replaced with something like ‘Hypotheses from Conventional assimilation ….’ With the topic and number of each hypothesis as a subheading of Heading 3. instead of as Headings 4. to 6. Also identify the relevant hypothesis in the Results Reporting. I think that would make this section clearer – and give readers a way to check back from the Results.

More conceptually, I see the hypotheses as grounded in Segmented Assimilation Theory rather than in both CST and SAT. This is not picked up at all in the Discussion, although a contrast is promised on p.3. Could the authors please check their intentions and make them explicit in their justification of the 9 hypotheses – by editing down the text of existing Sections 4 to 6 and not adding text.

  1. The Argument. While the authors have been careful to document their argument, I think the Abstract and the last paragraph of the paper could reflect their major findings a little more strongly. The last sentence seems lame, given the findings they are claiming and seeing as relevant. The sentence “Nevertheless, existing scholarship….” in the Abstract seems a bit general and vague. Again are the authors seeing CST and SAT as equally important to their analyses and findings?

  1. An important concept to clarify: “resilient mental forces”

The authors refer to a concept they measure as “resilient mental forces”. It is central to their argument. But it is measured as a second order variable including Rosenberg self-esteem ratings and students’ educational expectations (bottom of p. 9). While the self-esteem ratings had good correlations between waves 1 and 2, the future orientation ratings were low (0.179 & 0.261 for over 3000 participants). The authors have a long justification of this construction. I accept their statistical construction (pp. 9, 10), but I am not convinced about that this variable is more useful than self-esteem and educational expectations (or orientations) as separate variables. I also think this variable tends to be over-represented as “resilient mental forces”. The argument at top of p. 10 does not really justify putting the self-esteem and educational expectations together conceptually. I do not expect a revision of all the statistical analyses, but I think the authors should consider how they label this constructed variable, and the inferences they draw about it in the Results and Discussion. It seems to be a crucial issue for the study and theoretically.

  1. A few concepts and expressions to be cleared up:
  2. 2. “we intended to construct a causal relationship”. Surely the intention was to construct a model of a set of possible causal relationships? The last para on p.2. mentions “to construct and delineate the causal effects” - this is more reasonable, although I would suggest a possible set of causual effects – given that the data are pretty limited.
  3. 3. “its universal truth” witgh reference to CST. What does that mean?
  4. 4. “Correspondently” Is that a word?
  5. In Table 2 there is a heading Mean (%) but the columns includes proportions not % ages. All tables include mixed material – categorical and numerical. Given that there will be 7 tables, the authors could consider marking off sections of the tables with subheads – presenting only crucial information and organizing the material with the reader in mind. I would have expected 2 decimal places in the tables.

Author Response

Dear Dr Zorana Petrovic,

Thank you for replying us with reviewers’ comments, now we have revised our manuscript titled “Where There’s a Will There’s a Way? Social or Mental Forces of Successful Adaptation of Immigrant Children in Young Adulthood(ijerph-1695363)” according to the comments of the reviewers. By the way, we are here thankful for the useful and insightful comments of the three reviewers, in which we can feel their careful and constructive contributions. We reply our responses below (as comments of the three reviewers may be related, we hope to give the whole reply to each reviewer commonly):

For reviewer 1

1)Now the whole paper has been proofread and revised to make the presentation more concise and coherent logically and consistently. However, please note that as this study involves comparatively complicated arguments, theoretical constructions, and hypotheses, and analyses, hence the contents of the manuscript must be not such straightforward and easily understandable.

2)We suggest putting Figure 1-4 in the part of Appendix, but request to remain Table 1-8 in the contents of the manuscript. This is because the Tables are directly related to the logics presented in the study, which are helpful for readers to understand the comparatively complicated logics and study relationships of the research.

3)Now the references have been reduced to make the citations more simple and less lengthy, in which we try to consistently keep recent publications as references in the study.

4)“Heading 3 (top of p.3)” is now changed to “2.1. Hypothesis of Assimilation Theories and Adaptation of Immigrant Children” in response to the suggestion of reviewer 1.

5)The Abstract is now largely edited to make the presentation more consistent, salient, and indicative of the main findings and contributions of the study.

6)The concept of “resilient mental forces” has been clearlynow defined and presented in the manuscript (please refer to First paragraph of page 2 and “2.3. Mental Forces and Adaptation of Immigrant Children”), and as suggested by reviewers we now generally use the term “mental forces” to present this latent concept throughout the paper.

7)For clearly presenting the study relationships of this research and the causal effects of social and mental forces of immigrant children in adolescence on their successful adaptation in young adulthood, now we revised the study purpose mentioned in page 2 to “In this study, we intended to construct a causal relationship investigating the effects of social and mental forces and their interplay on immigrant children’s successful adaptation in terms of college graduation, enrolment in postgraduate study, and first and current occupational attainments from early adolescence to young adulthood”, and Hypothesis 1 to 9 are now more accurately responsive to the above study purpose.

7)Now for presentation of “its universal truth” witgh reference to CST.”, we changed the whole sentence to “Despite its effective predictive power of development of immigrant children, however some researchers have recently challenged its assertion of variegated and segmented assimilations”, which we thought can more clearly and accurately deliver the concept that we want to present.

8)The word “Correspondently” has been deleted, and the sentence is now directly presented “Research has showed that higher self-esteem and positive future expectation are related to better adaptation of youths.”

9)Now for Table 2, % is replaced by ‘Proportion’, and more information is added in the endnote of Table 2, which include “Mean is for continuous variables and proportion is for dummy or dichotomous variables.” As we mentioned above that Table 1-8 are used to present readers and let them understand the comparatively complicated study relationships, hypotheses, nature of data used, and findings of modeling analyses of the current research, hence they are not superfluous. As the regression coefficients and numerical results are presented in 3 decimal places throughout the whole paper for accuracy to let researchers and readers more clearly grasp the findings, hence we request to remain the presentation of Tables in 3 decimal places.

For reviewer 2

1)We do not try to confuse readers regarding the United States and America. In the whole study we now consistently presented the data, immigrant children, and their families coming from the United States, and the word that involves “America” in the a sentence is now rewritten to “These two assimilation theories highlight the significance of social mobility and convergence into the mainstream culture of host societies as the final end, resonant with what Gordon [20] claimed as the white Protestant middle-class in the United States of America.”

2)For the title of “Where There’s a Will There’s a Way? Social and Mental Forces of Successful Adaptation of Immigrant Children in Young Adulthood”, we hope to remain the phrase of “Where There’s a Will There’s a Way?” as it can highlight the main study intention and purpose of the current research to investigate and compare the effects of social forces and mental forces on adaptation of immigrant children in young adulthood. For this the phrase of “Where There’s a Will There’s a Way?” contains a “?” to let readers and researchers consider their impacts in a comparative and thoughtful sense.

3)For the suggestion to concern about “or Inclusive Education, Multiculturality Programmes”, we now highlighted this concern in the Discussion to mention the data of CILS collected in the early 1990s and early 2000s, in which provisions of interventions and programmes to help the adaptation of immigrant children were uncommon and no such information collected in the CILS data. This is the limitation of the current study, which now presents in the last paragraph of the Discussion that “In addition, the data of CILS were collected in the 1990s and early 2000s, during which due to the dominant views of conventional assimilation and segmented assimilation theories and endorsement of economic liberalism, other information such as intervention programmes of inclusive education and multicultural competence services for equalizing development of immigrant children are not available. For this, immigration researchers should incorporate data of intervention services or policy supports, if available, as moderators or mediators when investigating the effects of social and mental forces on long-term adaptation of immigrant children. In sum, immigration research scholars in the future are suggested to include not only the effects of social and mental forces as well as their interactive influences on adaptation of immigrant children, but also need to adopt a more culturally and temporally diverse research design for clearly examining the complicated processes of immigrant children’s adaptation and life trajectories.”

4)For “lack of Problem and Objetives setting”, the manuscript in fact has mentioned its study purpose and contributions in the second paragraph of Introduction as below:

“In this study, we intended to construct a causal relationship investigating the effects of social and mental forces and their interplay on immigrant children’s successful adaptation in terms of college graduation, enrolment in postgraduate study, and first and current occupational attainments from early adolescence to young adulthood. The present investigation surpasses prior immigration research. First, it covers an extended developmental period of immigrant children from early adolescence to young adulthood stretching a 10-year time span and concurrently examines multiple adaptation outcomes. Second, the effects of social forces in comparison to that of mental forces on successful adaptation of immigrant children are empirically examined to broaden our knowledge on the developmental trajectories of this population. Third, due to social and cultural segregation in different ethnicities, this study decomposes the clustering nature of their socialization contexts by concomitantly investigating both individual- and contextual-level influences of social and mental forces on immigrant children’s adaptation outcomes in young adulthood [13]. Finally, and most important, this study analyzes the interplay between social and mental forces to explore their possible dynamic and conditional impacts on successful adaptation of immigrant children in young adulthood.”

5)For including more structural predictors in the study, as mentioned above, the data of CILS was collected in the 1990s and early 2000s, during which the predominant view of conventional assimilation and segmented assimilation theories prevailed and now these two assimilation theories are still strongly regarded by immigration researchers. For this, we have tried to include those most important and pertinent structural predictor variables related to these two assimilation theories as more as we can while at the same time considering the lengthy contents of the study. Hence, a necessary balance is needed to reach for responding to the main study purpose of the current research. We hope the reviewer understand this constraint.

6)For consideration of current European immigration research, we have now mentioned the limitations of the current research in the part of Discussion due to the data of CILS, and also suggest researchers to adopt more innovative and insightful research approaches to study immigrant children in the future.   

For reviewer 3

1)For “attainment s” at line 12 of the abstract, now it is corrected.

2) For “the profound implications of a successful integration into host societies by immigrants and their children at lines 6-7 of your Introduction in a numbered list. Example: (i) legal order, (ii) social stability, (iii) economic and…, etc.,”, now we presented as “Specifically, successful integration to host societies by immigrants and their children, a process referring to economic mobility and social inclusion, has profound implications for Western immigrant-receiving countries, such as legal order, social stability, economic and labor development, cultural diversity, and general societal well-being [3, 4].” This presentation we consider as more corresponding to the presentation style of scholarly study involving social factors.

3) For “family SES (socio-economical statuses)”—first time it appears at line 45 page 3 of your paper, we now changed to “family socio-economic status (SES)” in the first place when this term appears in the paper, and then we use family SES in the later presentations throughout the paper.

4) For “Are your insights into the different structural and societal factors in relation to the adaptation of immigrant children restrained to Western capitalist host societies?”, this research intends to investigate and compare the effects of social forces and mental forces of immigrant children in adolescence on their later successful adaptation in young adulthood, social forces of segmented assimilation theory (family composition, family SES, and reception mode) are expected to be strongly influential on adaptation of immigrant children. In fact, family socialization and the impacts of proximal environment are most direct and robust socialization agents contributive to development of children, especially in capitalist societies where emphasize more on the resource-driven social selectivity process of education and development, although other less capitalist regions still regard the importance of family and receiving environment in influencing human development (please refer to the first paragraph of page 19 in Discussion).

5) For “Check “Hypothesis 5” first time it is mentioned in the middle of page 6 is in capital letter instead of the other hypotheses along the paper are in small letter”, now Hypothesis 5 is corrected to “We expect that the effects of mental forces on college graduation would be more salient for immigrant children from non-two-parent family, lower family SES, and negative reception conditions as compared to their counterparts of two-parent family, higher family SES, and positive/neutral reception status (Hypothesis 5).”

6) For “Is “incorporation mode,” the same as “assimilation/adaptation mode”?”, now we have clarified their same meaning, in which we writes “Accordingly, segmented assimilation theorists deem that family composition, family socioeconomic status (SES), and mode of incorporation, or called reception mode, are the most critical structural contributors relating to successful adaptation of immigrant children [2, 3, 25].” (please refer to the second paragraph of page 3).

7) For “After hypothesis 8, the discussion is redundant. Please check and incorporate only part of the text it in the previous hypotheses. Please delete hypothesis 9 contravenes hypothesis 8”, we do not agree that Hypothesis 8 and Hypothesis 9 are contradictory with each other, in which hypothesis 8 expects mental forces would have a stronger effect on their current occupation attainment for immigrant children in the contexts of non-two-parent family and lower family SES conditions compared to their peers in the contexts of two-parent family and higher family SES conditions after their accumulating certain working experiences and social capitals (as immigrant children in the contexts of non-two-parent family and lower family SES conditions would not like immigrant children in the context of negative reception to be prejudiced). However, hypothesis 9 posits that immigrant children of negative reception would continuously keep biased in the labor market, hence their personal effort and resilience would be less effective to help immigrant children of negative reception outperform their counterparts of positive/neutral reception, although the effect of mental forces is still believed to have a significant effect on current occupation attainment of immigrant children. For this, the effects of mental forces would be confound to be indistinguishable between immigrant children in the contexts of negative reception and positive/ neutral reception. Accordingly, our argument in the manuscript as

“However, human and social capital and work experience accumulate incessantly and persistently after graduation and hence the opportunity of first occupation would help immigrant children in non-two-parent family and lower family SES contexts compensate for prior family and social disadvantages [56, 57]. Thus, we expect that the effects of mental forces on current occupational attainment would be more salient for immigrant children in the contexts of non-two-parent family and lower family SES conditions than do their counterparts of two-parent family and higher family SES status (hypothesis 8). However, due to negative reception derived from social prejudices and ideological biases entrenching with deep cultural and historical roots [3, 15], accumulation of human and social capital and work experience may not easily help immigrant children of negative reception to eliminate their unfavorable treatment in the labor market, a phenomenon we call as persistent countervailing effects of cultural and historical origin. Therefore, we expect that the effects of mental forces on current occupational attainment would still be confounded to be indistinguishable among immigrant children in the negative and positive/neutral reception contexts (hypothesis 9). Table 1 summarizes the dynamic and conditional effects of immigrant children’s mental forces on different adaptation outcomes across the contexts of family composition, family SES, and incorporation mode, in which H connotes the higher effects of immigrant children’s mental forces in specific contexts of family composition, family SES, and incorporation mode compared to their counterparts in the opposite contexts, where L connotes the lower effects of mental forces. If comparable effects of immigrant children’s mental forces in the two relative contexts are expected, N is used to connote their parallels.” (Please read second paragraph of page 7).

Please note that the above argument does not mean mental forces insignificant, in contrast mental forces of immigrant children are proved to be an important predictor of immigrant children’s college graduation, engagement in postgraduate study, first and second occupation attainments.

8)For “Please check numbers at page 8 and put them with decimal point instead of commas. Example: 5.262 immigrant youth respondents; 3.613 respondents; another example is in the title of Table 2 at page 12 N=3.344, and so on”, the numbers presented in the manuscript are correct, in which N=5,262 and n=3,612 mean the numbers of immigrant youth participants in the wave-1 and wave-3 survey of CILS.

9) For “Please re-structure pages 1-11 of your paper because the discussion is too long”, the whole paper is now re-structured and proofread for more accurate, coherent, and concise, and the discussion is further improved. However, as responses to reviewer 1 and 2, the current study involves the construction of complicated study relationships to investigate and compare the effects of social and mental forces on successful adaptation of immigrant children from early adolescence to young adulthood. Hence, the presentation would be possibly less simple than other common research published. We hope the Editors and Reviewers understand.

Reviewer 2 Report

English writing: You can not write America instead of USA. This is a wrong way to use the name of a counetry as the name of the whole continent.

I do not recomend to use There´s, better There is.

You have to compare Assimilation Perspective to others as Segregation or Inclusive Education, Multiculturality Programmes.

I disagree strongly with the concept of "intact family" when the father and the mother are present. This is the patriarchal model o f family only and this  is an opresive model of family. So, you need to explain this concept.

There is a lack of Problem and Objetives setting. No tools are explained and why you have selected them.

Strcutural predictors are poor and they are not enough fo this research.

The review is not interesting as a model for inmigrants. You should consideser some really innovative experiences developed ni European countries.

Author Response

(The authors gave the same response as above.)

Reviewer 3 Report

This article provides insights into the different structural and societal factors in relation to the adaptation of immigrant children in Western capitalist host societies, termed as “social” or “mental forces” in this paper, in relation to immigrant children’s successful adaptation in adolescence. To this aim the authors mean to examine and compare both the effects of social and mental forces measured in adolescence of immigrant children on their multiple adaptation outcomes in terms of college graduation, engagement in postgraduate study, first and current job attainments in young adulthood with a Bayesian multilevel modeling framework. The authors expect themselves that family composition, family SES, and incorporation mode specified by segmented assimilation theory would be simultaneously predictive of immigrant children’s successful adaptation in young adulthood at both individual and school levels.

The authors’ main employed method is the self-construction of the causal effects of social and mental forces and their interplay on successful adaptation of immigrant children in young adulthood by commenting data of a large and representative sample from the Children of Immigrants Longitudinal Study (CILS), and through a multilevel model due to large sample size. See abstract, lines 7-19. See also Section 2 at the end of page 2, and starting Section 7 at page 8.

Authors main hypotheses are the following:

  • h0: no relationship between social and mental forces (also latent) measured in adolescence of immigrant children on their multiple adaptation outcomes (such as: a. college graduation, b. engagement in postgraduate study, c. first and current job attainments in young adulthood) and successful adaptation of immigrant children in young adulthood,
  • h1: social and mental forces measured in adolescence of immigrant children on their multiple adaptation outcomes in terms of college graduation, engagement in postgraduate study, first and current job attainments in young adulthood interplay on successful adaptation of immigrant children in young adulthood,
  • h2: (i) family composition, (ii) family SES (socio-economical statuses), and (iii) incorporation mode specified by segmented assimilation theory are simultaneously predictive of immigrant children’s successful adaptation in young adulthood at both individual and school levels,
  • h3: the resilient mental forces of immigrant children formed by their self-esteem and educational future expectation that are treated as a second-order latent construct are positively and robustly predictive of their successful adaptation later in young adulthood even adjusting for the structural effects proposed by conventional assimilation theory and segmented assimilation perspective,
  • h4: the effects of resilient mental forces of immigrant children on their successful adaptation are significantly stronger than the structural effects of family composition, family SES, and incorporation mode informed by segmented assimilation theory,
  • h5: the effects of resilient mental forces on college graduation are more salient for immigrant children from non-intact family, lower family SES, and negative reception conditions as compared to their counterparts of intact family, higher family SES, and positive/neutral reception status,
  • h6: the effects of resilient mental forces of immigrant children on their engagement in postgraduate study are more salient for immigrant children in non-intact family, lower family SES, and negative reception conditions than for their counterparts in intact family, higher family SES, and positive/neutral reception status,
  • h7: the effects of resilient mental forces on first occupational attainment are indistinguishable among immigrant children in the different contexts of family composition, family SES, and incorporation mode,
  • h8: the effects of resilient mental forces on current occupational attainment are more salient for immigrant children in the contexts of non-intact family and lower family SES conditions than do their counterparts of intact family and higher family SES status.

In their Section 1. Introduction, the authors enhance the reemergence of research interest in immigrants and the future of their offspring, especially as spurred by the contemporary global refugee crisis. The authors have inquired into the existing literature examples of the implications a successful integration into host societies by immigrants and their children may have, such as:

  • legal order,
  • social stability,
  • economic and labor development,
  • cultural diversity, and
  • general societal well-being of immigrant-receiving countries.

The authors envisage to investigate both individual- and contextual-level influences of social and mental forces on immigrant children’s adaptation outcomes in young adulthood. Beyond a conventional assimilation perspective and/or the dominant theory of segmented assimilation, they define “social and mental forces” as «the human mental strengths and psychological determination of immigrant youth to achieve a successful adaptation, such as successful college graduation, engagement in postgraduate study, first and current occupational attainments.» The authors mean to investigate these factors to somehow extend the developmental period of immigrant children from early adolescence to young adulthood stretching a 10-year time span and concurrently examine multiple adaptation outcomes.

In their Section 2. Theoretical Framework of Successful Adaptation of Immigrant Children, the authors shortly describe the general situation of million of children under 18 live in the United States and are from immigrant families. Since notwithstanding comparatively disadvantaged and difficult conditions, some of them are capable of attaining higher academic and social achievement, the authors wish to investigate how and under what circumstances immigrant children can successfully adapt in young adulthood. See page 2. Rumbaut’s sociology on immigrant children’s successful adaptation in young adulthood is mentioned.

In their Section 3. Conventional Assimilation Theory and Segmented Assimilation Perspective on Adaptation of Immigrant Children, the authors discuss the topic’s approach by both the conventional assimilation theory and/or the segmented assimilation perspective in the existing literature. Both belong to a structuralist and functionalist tradition, though the first praises for a more linear upward assimilation by the side of immigrants’ youths, while the latter emphasizes ethnic divergences in the process of assimilation as well as “several distinct paths of adaptation,” as stressed among others by Haller, W.; Portes, A.; Lynch, S. M. (2011) and Portes, A.; Fernandez-Kelly, P.; Haller, W. (2009).

In their Section 4. School Context and Adaptation of Immigrant Children, the authors discuss immigrant children’s behaviors through interaction with classmates, teachers, and other school personnel as well as observations of school environment and climate. The authors then try to analyze both individual- and school-level effects of family composition, family SES, and incorporation mode highlighted by segmented assimilation theory on immigrant children’s adaptation outcomes of college graduation, enrolment in postgraduate study, first and current occupational attainments in young adulthood.

Section 5. Mental Forces and Adaptation of Immigrant Children, is a summary of the commonest characteristics at human development, such as (i) self-esteem (sub-categorized into a. self-worth; b. self-acceptance; c. confidence in forming a crucial cognitive basis to affect immigrant children’s values, goals, and behavioral choices); and (ii) a sense of hopeful future (sub-categorized into a. ability to plan for the future, and b. ability to set actions to attain planned goals).

These two characteristics are somehow overlapping as shown by the same authors through the mentioning of the relevant literature such as Verdugo et al. (2018).

Section 6. Context-driven Resilient Response of Immigrant Children’s Mental Forces and Their Adaptation, is an introductory explanation of the main perspective findings (h5) of the authors: that is, “the effects of resilient mental forces on college graduation” are “more salient for immigrant children from non-intact family, lower family SES, and negative reception conditions as compared to their counterparts of intact family, higher family SES, and positive/neutral reception status.” Same is deemed by authors on immigrant children’s engagement in postgraduate study (h6). See at the end of page 6. Instead, since the labor market is different from the realm of higher education, the authors in this case think immigrant children in non-intact family, lower family SES, and negative reception conditions have relatively weak family resources and social connections and therefore are in a less favorable position to have a better first occupation attainment as compared to their counterparts of intact family, higher family SES, and positive/neutral reception. So that the effects of resilient mental forces on first occupational attainment are indistinguishable among immigrant children in the different contexts of family composition, family SES, and incorporation mode.

Another change is expected in the case of the immigrant children’s current occupational attainment, because after first work experiences, they will try to compensate for prior family and social disadvantages. Then, as hypothesis n. 8 the authors expect the effects of resilient mental forces will be more salient for immigrant children coming from disadvantaged contexts respect to their counterparts in privileged families with higher SES.

Table 1 at page 7 is a self-attribution by authors through a L-H-N coding of the family SES characteristics of immigrant children and their multiple adaptation outcomes in terms of college graduation, engagement in postgraduate study, first and current job attainments in young adulthood.

Section 7. Data and Methods and its sub-paragraph 7.1. Sample, shows the characteristics of the secondary sample of 3.344 respondents drawn by CILS data, one of the largest longitudinal studies on the life course development of immigrant children in the United States. Time span of the CILS panel has been 10 years, from 1992 until 2002. Original sampling by CILS was restricted to only two immigrant-receiving regions of the US and non-Hispanic White youths were not included in the sample. As strong advantage, CILS is a large panel sample of immigrant children from 77 nationalities and contains fruitful and useful individual, family and school variables that pertain to their adaptation and development in young adulthood. Sub-paragraphs 7.2. Measures, and 7.2.1. Outcome Variables of Successful Adaptation, show how graduate and/or professional degree of post-graduate schools are measured through CILS’ dichotomous variable (1=yes, 0=no). First and current occupational attainments are measured though CILS’ Treiman job prestige scores with a score ranging from 0 to 100 (where higher scores mean better occupational achievement).  Sub-paragraph 7.2.2. Structural Predictors of Segmented Assimilation Theory, shows how family composition, family SES, and incorporation mode are obtained through wave-3 survey of CILS. Obtained data drawn from CILS are then a mix of dichotomous variable, unit-weighted standardized scale of parents’ immigrant children’s characteristics, and average of other features such as individual-level family SES across immigrant youth students in the same school. Sub-paragraph 7.2.3. Structural Predictors of Conventional Assimilation Theory, illustrates a mix of other measurements in the panel such as generations of immigrant children, length of residence, English use, and interaction with native friends adopted from wave-1 data of CILS. Sub-paragraph 7.2.4. Resilient Mental Forces of Immigrant Children, illustrates:

  • first-order latent factors’ measurements, that is (i) self-esteem and (ii) positive educational expectation. Rosenberg self-esteem scale (RSES) was used to measure immigrant children’s self-concept, while educational expectation was measured by combing two questions asking immigrant children the highest education attainment they hoped to achieve and could be attained realistically.
  • Other EFA (exploratory factor analysis) such as oblique rotation and ML (maximum likelihhod) factors for large datasets (with Bartlett’s Test of Sphericity) are employed to construct second-order latent predictors of resilient mental forces of immigrant youths.

Then both have been merged to form resilient mental forces of immigrant children.

Sub-paragraph 7.2.5. Individual-Level Control Variables, shows the control variables have been considered by authors in their secondary sample, such as (i) gender, (ii) age, (iii) siblings, (iv) total mathematics and reading scores of standardized Stanford Achievement Tests (SAT10), and (v) ethnic origin of immigrant children measured in wave-1 survey of CILS. Sub-paragraph 7.2.6. School-Level Control Variables, shows school-level control variables such as (i) school location, (ii) school type, (iii) minority-receiving school status, and (iv) school academic achievement through aggregated standardized test scores of the school.

Sub-paragraph 7.3. Modeling Procedures, describes the Bayesian analysis approach to obtain more reliable parameter estimates and preclude an inflation of Type I errors commonly found in ML. A detailed explanation of the Bayesian modeling it follows until page 12.

Section 8. Results, shows percentages of outcomes in the sample. Table 2 at page 12 shows demographics of Immigrant Children in CILS’ drawn sample. In Table 3 and Table 4 at pages 13-14 the authors calculate the ICC (Intraclass correlation) for the school and occupational attainments of immigrant children, and other centering within context (CWC) parameters may account for unobserved time-varying factors that affect the adaptation of immigrant children in Western capitalist host societies. The authors’ generalized linear mixed model (GLMM) is then exemplified in Table 5 at page 15 where first-order loadings and second-order loadings are shown for 4 classes of immigrant children’s study and occupational achievements. Hypotheses about effects of social and mental forces both at on adaptation outcomes of immigrant children in young adulthood are tested in Table 6 at page 16 at both individual and school levels. Wald tests of parameter constraints (see Table 7 at page 17) by setting equivalence of the aforementioned effects on immigrant children’s college graduation, engagement in postgraduate study, first and current occupational attainments confirms authors’ h5, h6, and h8 before mentioned, so doing letting avoided the “generalized stigma and insecurity of illegal status” of immigrant children is generally set forth in these kinds of studies. Table 8 at page 18 concludes with Z-values the authors’ discussion with the enhancing of second-order latent construct (that is the resilient mental forces) are positively and robustly predictive of immigrant children’s successful adaptation later in young adulthood.

Section 9. Discussion, and Figures 1-4 at pages 20-21 show the statistical posterior predictive checks through graphical inspection of the operationalization of CILS variables by authors involves various losses of information, as immigrant children can be heterogeneous regarding migration generation and the more than 70 countries of origin. 

CHANGE REQUEST:

  1. Check the word “attainment s” at line 12 of the abstract,
  2. Put the profound implications of a successful integration into host societies by immigrants and their children at lines 6-7 of your Introduction in a numbered list. Example: (i) legal order, (ii) social stability, (iii) economic and…, etc.,
  3. Specify the acronym at SES— “family SES (socio-economical statuses)”—first time it appears at line 45 page 3 of your paper,
  4. Are your insights into the different structural and societal factors in relation to the adaptation of immigrant children restrained to Western capitalist host societies?

Please check and revise as appropriate,

  1. Check “Hypothesis 5” first time it is mentioned in the middle of page 6 is in capital letter instead of the other hypotheses along the paper are in small letter,
  2. Is “incorporation mode,” the same as “assimilation/adaptation mode”? Please check and corroborate the use of this meaning as appropriate,
  3. After hypothesis 8, the discussion is redundant. Please check and incorporate only part of the text it in the previous hypotheses. Please delete hypothesis 9 contravenes hypothesis 8.

Make smoother all your discussion lessening text here and there in your 1-6 paragraphs,

  1. Please check numbers at page 8 and put them with decimal point instead of commas. Example: 5.262 immigrant youth respondents; 3.613 respondents; another example is in the title of Table 2 at page 12 N=3.344, and so on,
  2. Please add the acronym of Rosenberg self-esteem scale (RSES) at the start of 7.2.4. paragraph, page 9.
  1. Please re-structure pages 1-11 of your paper because the discussion is too long. The very interesting annotations about CILS data treatment in your secondary sample may be blurred from this very long text. I advise for the reading of Kassis, W. et al. (2021) paper, available from: https://www.sciencedirect.com/science/article/pii/S0883035521001324.

With Kind Regards,

References:

Feaster, D., Brincks, A., Robbins, M., & Szapocznik, J. (2011). Multilevel models to identify contextual effects on individual group member outcomes: a family example. Family process, 50(2), 167–183. https://doi.org/10.1111/j.1545-5300.2011.01353.x

Garmezy, N. (1993). Children in Poverty: Resilience Despite Risk. Psychiatry, 56(1), 127–136. doi: 10.1080/00332747.1993.11024627

Garmezy, N. (1993). Vulnerability and resilience. In D. C. Funder, R. D. Parke, C. Tomlinson-Keasey, & K. Widaman (Eds.), Studying lives through time: Personality and development (pp. 377–398). American Psychological Association. https://doi.org/10.1037/10127-032

Hao, L., & Woo, HS. (2012). Distinct trajectories in the transition to adulthood: are children of immigrants advantaged?. Child development, 83(5), 1623–1639. https://doi.org/10.1111/j.1467-8624.2012.01798.x

Kasinitz P., Battle J., Miyares I. Fade to Black? The Children of West Indian Immigrants in South Florida. In: Rumbaut RG, Portes A, editors. Ethnicities: Children of Immigrants in America. Berkeley, CA: University of California Press and Russell Sage Foundation; 2001. pp. 267–300.

Kassis, W.; Govaris, C.; Chouvati, R.; Sidler, P.; Janousch, C. ; Ertanir, B. (2021). Identification and comparison of school well-being patterns of migrant and native lower secondary-school students in Greece and Switzerland: A multigroup latent profile analysis approach. International Journal of Educational Research, 110. https://doi.org/10.1016/j.ijer.2021.101863

Portes A., Rumbaut RG. Legacies: The Story of the Immigrant Second Generation. Berkeley, CA: University of California Press and Russell Sage Foundation; 2001.

Worswick, C. (2004). Adaptation and Inequality: Children of Immigrants in Canadian Schools. The Canadian Journal of Economics / Revue canadienne d'Economique, 37(1), Feb., pp. 53‒77.

Author Response

(The authors gave the same response as above.)

Round 2

Reviewer 2 Report

I agree now with the changes and improvement

Author Response

Dear Dr Robert Zhao,

We have revised the manuscript according to the editor’s and reviewer’s comments, and our responses are below:

#For reviewer 2

-I agree now with the changes and improvement

Reply: Thank you.

#For the editor’s comments

Thank you for the revision. Overall, the manuscript is well written and well addressed reviewers' comments and feedback. I have a minor suggestion for authors:
# The abstract needs to be reorganized:
- Please delete the word "so-called" from the abstract.

Reply: we have deleted "so-called” from the abstract.
- Please mention the sample sizes in the abstract.

Reply: Now the sample size and the source of the data are reported in abstract.
- Please also report the findings with OR and/or β value, and so on.

Reply: Related findings of ORs and βs have reported in abstract now.

Finally, we are thankful for the effective and constructive reviewing processes of the editorial team of IJERPH, and also grateful to the meaningful and useful comments of the reviewers.

Thank you for your attention.

Best

Jerf Yeung